# DOES DIFFUSION BEAT GAN IN IMAGE SUPER RESOLUTION?

## ABSTRACT

There is a prevalent opinion [1] that diffusion-based models outperform GAN-based counterparts in the Image Super Resolution (ISR) problem. However, in most studies, diffusion-based ISR models employ larger networks and are trained longer than the GAN baselines. This raises the question of whether the high performance stems from the superiority of the diffusion paradigm or if it is a consequence of the increased scale and the greater computational resources of the contemporary studies. In our work, we thoroughly compare diffusion-based and GAN-based super resolution models under controlled settings, with both approaches having matched architecture, model and dataset sizes, and computational budget. We show that a GAN-based model can achieve results comparable or superior to a diffusion-based model. Additionally, we explore the impact of popular design choices, such as text conditioning and augmentation on the performance of ISR models, showcasing their effect in several downstream tasks.

## 1 INTRODUCTION

In recent years, the field of Image Super Resolution (ISR) has witnessed significant advancements, primarily attributed to improvements in image generation frameworks, mostly Generative Adversarial Networks (GANs) Goodfellow et al. (2014) and Denoising Diffusion Models (DDMs) Ho et al. (2020).

GANs introduced the adversarial training framework, which led to their remarkable ability to fit and recover narrow, unimodal distributions while struggling with complex multi-modal data. This trade-off allowed GANs to dominate the area of ISR for half a decade Ledig et al. (2017b); Wang et al. (2018); Karras et al. (2017); Wang et al. (2021), pushing the boundaries of resulting image quality, especially for high upsampling factors.

In contrast, more contemporary DDMs provide a robust framework for image generation, reliably covering multi-modal data distributions. Yet, the diffusion ability to yield high-quality and diverse outputs comes with a cost of high computational load during training and inference.

At the moment, the literature suggests that diffusion models have become a dominant source of state-of-the-art methods in generative modeling, including ISR (Saharia et al. (2021); Li et al. (2021); Yue et al. (2023); Yu et al. (2024)). However, a comprehensive comparison between GAN and diffusion models in the context of ISR under controlled conditions still needs to be explored. In particular, GAN-based methods that modern state-of-the-art methods are compared with are typically represented by Real-ESRGAN Wang et al. (2021), which has orders of magnitude fewer parameters, training data samples, and computational budget.

This work aims to thoroughly compare GAN and DDM frameworks for Image Super Resolution. Given the unique strengths of both GANs and diffusion models, understanding their comparative performance in ISR tasks is crucial. This comparison requires a rigorous and controlled experimental setup to ensure comparable protocol, *i.e.*, standardized variables such as the size of the training dataset, model complexity, and controlled allocation of computational resources.

Furthermore, we explore the influence of various design choices on the performance and generalization of these models. In particular, we investigate the value of practically useful Real-ESRGAN

---

[1]See, for example, (Moser et al., 2024).

Wang et al. (2021) augmentations and validate the necessity of incorporating textual conditioning (*e.g.*, image captions). The contributions of our work are the following:

– We thoroughly compare GAN and DDM in Image Super Resolution problem: if scaled appropriately, GAN-based models surpass quality of diffusion-based ones in classical SR and yield similar quality in SR in the Wild.

– We investigate the influence of textual conditioning in the form of non-ISR-specific image captions and find that it has no significant effect on both frameworks.

– We explore the pros and cons of a complex augmentations pipeline for each paradigm and reveal that it may introduce some quality benefits but comes with its cost, especially hurting DDM training dynamics.

## 2 RELATED WORK

**GANs for SR**    Shortly after the introduction of the original GAN paper Goodfellow et al. (2014) for the general image generation problem, C. Ledig *et al.* presented SRGAN Ledig et al. (2017b), which introduced perceptual loss, combining content loss (using a pre-trained VGG network Johnson et al. (2016)) with adversarial loss to generate high-resolution images with improved perceptual quality. The following research updated the initial scheme with improved architecture and feature extraction (ESRGAN Wang et al. (2018), ESRGAN+ Rakotonirina & Rasoanaivo (2020), LAPGAN Denton et al. (2015), LSMGAN Mahapatra et al. (2017)), more effective training mechanism (RankSRGAN Zhang et al. (2021b), PGGAN Karras et al. (2017)) more robust discriminator (RaGAN Jolicoeur-Martineau (2018)) or multiple discriminator models (MPDGAN Lee et al. (2019)), and sophisticated data pre-processing strategies to increase robustness to in the wild data (BSRGAN Zhang et al. (2021a), RealSR Ji et al. (2020), Real-ESRGAN Wang et al. (2021)). These methods built a strong foundation for GANs, which have become the de facto standard framework for all kinds of ISR problems.

Recent advances in DDMs resulted in ISR models with stunning restoration quality. This remarkable progress is at least partially attributed to scaling up models, data, and computational resources. While novel GAN-based methods make progress in catching up with updating and scaling (Giga-GAN Kang et al. (2023)), small-scale Real-ESRGAN remains the most commonly used baseline model in modern ISR research.

**Diffusion models for SR**    Recent developments in training and scaling of DDMs resulted in diffusion surpassing GANs for the general image generation problem Dhariwal & Nichol (2021) and several new methods aimed to transfer this success to the ISR problem. The pioneering ISR DDM approaches trained image-conditioned DDM from scratch, concatenating the input noise with either the upscaled version of a low-resolution image (SR3 Saharia et al. (2021)) or its hidden representation produced by a convolutional network (SRDiff Li et al. (2021)). Their impressive results relegated ISR GANs to the background and inspired the next-generation methods that propose to directly sample from the distribution of LR-images rather than the conditioned Gaussian noise ($I^2SB$ Liu et al. (2023a), ResShift Yue et al. (2023)). Another branch of research aims to incorporate generative prior of Latent Diffusion Models Rombach et al. (2022) by freezing their generative backbones and training only lightweight ControlNet-based Zhang et al. (2023) modules (StableSR Wang et al. (2023b), DiffBIR Lin et al. (2024), SUPIR Yu et al. (2024)). The latter class of methods typically uses additional condition sources, such as depth maps, segmentation maps, and text.

**Text-conditioned Super Resolution**    Most papers Yang et al. (2023); Wu et al. (2023) generally report that using additional conditions in ISR DDMs improves the resulting image quality. However, the case of text conditioning is more nuanced. Pixel-based natural images-focused ISR models (SR3 Saharia et al. (2021), SR3+ Sahak et al. (2023)) do not use textual conditioning. At the same time, several ISR models designed for upscaling inside cascaded diffusion systems, namely, Imagen Saharia et al. (2022), DeepFloyd Shonenkov et al. (2023), Kandinsky 3.0 Arkhipkin et al. (2023) in the $1024 \rightarrow 4096$ upscaler, incorporate textual information via cross-attention. In contrast, a more recent YaART Kastryulin et al. (2024) reports the lack of noticeable quality improvement and removes cross-attention at this stage. Latent diffusion-based ISR models also differ in this regard. StableSR Wang et al. (2023b) adopts the null-text prompt, effectively removing the text conditioning, and DiffBIR Lin et al. (2024) does not have any text conditioning. However, more recent SUPIR Yu et al. (2024) uses a dedicated multi-modal large language model to automatically caption input

images with non-ISR specific captions, reporting a noticeable performance boost. SeeSR Wu et al. (2023) goes further and proposes a method to generate ISR-specific captions to boost the model's generative ability, claiming its effectiveness in upscaling small objects with ambiguous semantics.

In this paper, we aim to shed light on the importance of textual conditioning for ISR in a practical setup when ISR-specific captions are not accessible, and the model has no intrinsic, generation-based prior to utilizing textual information.

**Variations of SR task**  A general SR problem can be formulated as follows. For a given high quality image $I^{HR}$, one applies some degradation operation $\mathcal{D}$ to produce low quality image $I^{LR}$ typically of smaller spatial resolution. This degradation can be a simple bicubic/area downsampling kernel or a more general degradation process. In the task of Blind Super Resolution (BSR) Liu et al. (2021), which has attracted significant attention recently, one aims to reconstruct low-resolution images exposed to *unknown* and *complex* degradations. Higher-order degradation models introduced in Wang et al. (2021); Zhang et al. (2021a) emulate possible types of corruptions that occur in the real world.

## 3 METHODOLOGY

**Models**  Since our goal is to compare GAN and diffusion Super Resolution under a comparable setup, we use the same architecture for both paradigms considered. Specifically, we adopt a network similar to $256 \times 256 \rightarrow 1024 \times 1024$ SR model from Imagen (Saharia et al. (2022)). It acts in the pixel domain and is based on Efficient U-Net architecture Ronneberger et al. (2015). This model has $600 - 700M$ trainable parameters, depending on the configuration. Details about architecture are provided in Appendix C.

GAN-based Super Resolution models take bicubic-upscaled low-resolution image and directly predict a high-resolution image, given a reference on training.

Diffusion Super Resolution models predict a noise applied to the high-resolution image, using the bicubic-upscaled low-resolution image as a condition. Following the prior work, the condition image is channel-wise concatenated with the noisy input.

Both models are almost identical in terms of parameters, but for the slight difference in the diffusion model having slightly more parameters due to the concatenation of noisy input and image condition in the input layer and timestep embeddings.

**Text conditioning**  One of the questions addressed in our work is the impact of text conditioning on the performance of Super Resolution models. In our work, we consider two types of text encoders: a CLIP-like (Radford et al., 2021) proprietary encoder transformer model that we call XL with 1.3B parameters and UMT5 (Chung et al., 2023) encoder-decoder with 3B parameters. The text caption, corresponding to the image, is first processed via the text encoder and then passed to the Super Resolution model via image-text cross-attention at the lowest resolution of the U-Net encoder, middle block, and decoder. This design choice follows Imagen (Saharia et al., 2022) work.

In our work, we study the impact of the prompts with semantic information *i. e.*, the ones that are typically available from large image-text datasets Schuhmann et al. (2022); Gadre et al. (2024), rather than image-quality aware captions adopted in SUPIR Yu et al. (2024) and SeeSR Wu et al. (2023). We believe it is timely to revisit the impact of these non-SR-specific texts due to their prominent availability in recently published image-text datasets Schuhmann et al. (2022); Gadre et al. (2024) used for training large-scale multi-modal models.

**Augmentation**  In our work, we study the impact of train-time augmentations in two setups: Super Resolution of down-scaled images and in case of a more general and complex degradation process. We train GAN and diffusion models with and without the image degradation model proposed in (Wang et al., 2021) to address both scenarios.

**Training**  Following the standard practice Ledig et al. (2017a); Wang et al. (2021); Saharia et al. (2021); Sahak et al. (2023), both types of models are trained on image crops. In the ablation study in Section 5, we also investigate the impact of finetuning on full-resolution images.

For GAN-based SR, we initially pretrain the generator model with $L_1$ loss only. This pretraining is essential since training from scratch with adversarial loss yields artifacts in agreement with prior work (Ledig et al., 2017a; Wang et al., 2018; 2021). Training with $L_1, L_2$ or any derivative training

objective alone leads to over-smoothed and blurry upscaled images. Adversarial training, introduced in SRGAN Ledig et al. (2017a), allows the production of images with sharp edges and distinguished high-frequency details. We adopt non-saturating adversarial loss for training without any additional regularization.

$$L_{D,\text{adv}} = \mathbb{E}_{I^{HR} \sim p(I^{HR})}[\log D_{\theta_D}(I^{HR})] + \mathbb{E}_{I^{LR} \sim p(I^{LR})}[\log(1 - D_{\theta_D}(G_{\theta_G}(I^{LR})))] \quad (1)$$

$$L_{G,\text{adv}} = -\mathbb{E}_{I^{LR} \sim p(I^{LR})}[\log D_{\theta_D}(G_{\theta_G}(I^{LR}))] \quad (2)$$

Above, $\theta_G$ and $\theta_D$ are the parameters of generator and discriminator, respectively. $I^{LR}$, $I^{HR}$ are the low-resolution and high-resolution images. We employ a U-Net with spectral normalization introduced in Wang et al. (2021) as a discriminator with twice as many hidden channels. Discriminator is trained from scratch.

During the $L_1$ pretraining stage, the generator model is only trained with $L_1$ loss. After that, we turn the adversarial loss on and reformulate the generator objective as a sum of $L_1$ loss and adversarial loss $L_{G,\text{adv}}$ with equal weights:

$$L_G = L_1 + L_{G,\text{adv}} \quad (3)$$

We note that usually GAN-based SR papers include an additional perceptual loss term, following SRGAN Ledig et al. (2017a) work. We experimented with the option of adding loss on VGG19 Simonyan & Zisserman (2014) features computed between the predicted and original high resolution image, but observed no improvement.

The diffusion model is trained on $\epsilon$-prediction objective with noise timesteps $t$ sampled uniformly from $[0, 1]$:

$$L_{\text{diff}} = \mathbb{E}_{I^{HR} \sim p(I^{HR}), I^{LR} \sim p(I^{LR}), \epsilon \sim \mathcal{N}(0,1), t \sim \mathcal{U}[0,1]} \|\epsilon - \epsilon_\theta(z_t, I^{LR}, t)\|_2^2, z_t = \alpha_t I^{HR} + \sigma_t \epsilon \quad (4)$$

Where $\alpha_t$ and $\sigma_t$ are determined by the noise schedule. We adopt standard variance-preserving (VP) schedule Ho et al. (2020) with linearly increasing variances.

**Inference**  GAN-based upscalers produce high-resolution images in a single forward pass, whereas diffusion models iteratively denoise input noise conditioned on a low-resolution image. For that, we adopt DPM-Solver++ Lu et al. (2023) as the one providing an excellent trade-off between generation quality and efficiency. We set the order of the solver to be two and chose the multistep method for sampling. All upscaled images are generated with 13 sampling steps. Our evaluations showed that a further increase in the number of sampling steps does not lead to visually noticeable image quality gains H.1.

## 4 EXPERIMENTS

### 4.1 EXPERIMENTAL SETUP

**Training Dataset**  We train our models on a large-scale proprietary dataset of 17 million text-image pairs. The dataset consists of $1024 \times 1024$ px images with exceptional image quality, high image-text relevance, and English captions. The high dataset quality is achieved by a rigorous filtering process described in detail in Appendix F.

We train our models on image crops to ensure training efficiency and sample variety. For that, we produce low-resolution - high-resolution (LR, HR) image pairs via extracting $256 \times 256$ px random crops of the original images, which are then downscaled to $64 \times 64$ px. For the finetuning stage, we interpolate 1024 px images to 256 px. `cv2.INTER_AREA` algorithm is adopted for image resizing.

**Training Hyperparameters**  We follow prior work in the choice of hyperparameters Wang et al. (2021); Sahak et al. (2023) and observe that the training with these choices is stable and converges fast enough. We fix the batch size for GAN and diffusion to ensure fairness of comparison on the number of samples seen during training and adjust the learning rate accordingly. Further increasing the learning rate with the goal of speeding up training leads to training instability and poor final results. For both models, the batch size is set to 512 for training on crops and 128 for full-resolution images. We adopt a constant linear rate with a linear warmup phase. The models are trained with Adam Kingma & Ba (2017) optimizer without weight decay. More details are provided in Appendix D.

While most of the related works adopt a predefined number of training iterations, we set the training duration dynamically and terminate the training only when the visual quality of the images upscaled by adjacent model checkpoints becomes indistinguishable from the humans' perspective. Since the changes become smaller throughout the training, we use an exponentially spaced grid with checkpoints from the following list of thousands of training iterations: [10, 20, 40, 60, 100, 140, 220, . . .], where we increase the interval between adjacent two checkpoints by a factor of two. Then, we estimate the human annotator preference using the Side-by-Side (SbS) evaluation for the current and previous training steps. If there is no improvement for three subsequent measurements, we conclude that the model performance is saturated and proceed to the next stage. We note that the training loss and full-reference metrics may continue to improve, but the improvement is imperceptible to human observers. In our work, user preferences are used both for early stopping and model evaluation. The SbS comparison setup is described and explained in greater detail in Appendix E.

**Evaluation Datasets**   We evaluate the performance of the SR models under study on the 244-images dataset constructed from the pictures from RealSR Ji et al. (2020), DRealSR Wei et al. (2020), and some additional high-quality samples from the web. LR 256 px images are obtained from HR images with `cv2.INTER_AREA`. Captions for images required as input for text-conditional models are generated with the LLaVA model Liu et al. (2023b).

To estimate the quality of image restoration under more complex and severe corruptions, we use the same dataset but with LR inputs obtained through Real-ESRGAN Wang et al. (2021) degradation pipeline.

In addition, we investigate the robustness of ISR models on out-of-distribution synthetic data in Appendix G. For that, we provide a qualitative comparison between GAN and diffusion models together with the baselines on the Super Resolution of downscaled images produced by the YaART Kastryulin et al. (2024) and SDXL Podell et al. (2024) models using prompts from DrawBench Saharia et al. (2022) and YaBasket Kastryulin et al. (2024) prompt sets.

**Baselines**   Our work's primary goal is to compare GAN and diffusion Super Resolution rather than establishing a new state-of-the-art in the domain. However, to demonstrate that our models yield quality competitive to the current state-of-the-art, we provide both qualitative and quantitative comparison with SUPIR Yu et al. (2024), DiffBIR Lin et al. (2024), RealESRGAN Wang et al. (2021), ResShift Yue et al. (2023).

## 4.2 Training Dynamics

**GANs converge faster than diffusion**   As discussed above, we train the model until the human annotators do not show a preference for the current training step over the previous one for three evaluations in a row. To have enough reference points, we train the models such that they cover at least six milestones—[10k, 20k, 40k, 60k, 100k, 140k] iterations.

We observe that GAN Super Resolution converges very quickly in all stages. After 40k steps of $L_1$ pretraining, the changes in the output of the SR model become imperceptible. Adversarial training converges relatively fast as well. After pretraining on crops for 40k steps, the performance of the Super Resolution model almost stabilizes. Prior works Wang et al. (2021); Zhang et al. (2021a); Liang et al. (2021) conduct training for 500k-1500k iterations with batch size $\sim$ 10 times smaller than the one used in our work. Therefore, the total number of samples seen is of the same order of magnitude as the referenced papers.

We note that in our experiments, in addition to worsening computational efficiency, pretraining on full-resolution images resulted in worse Super Resolution quality starting from the first thousands of training iterations. After several attempts to adjust training hyperparameters, we stopped this line of experimentation.

GANs are claimed to be prone to mode collapse Roth et al. (2017) and notoriously hard to optimize, requiring careful tuning and regularization for successful training. However, we did not encounter any difficulties with optimization. After several hundreds of iterations, our GAN models achieved equilibrium between the generator and the discriminator and continued to improve steadily until convergence.

Conversely, diffusion models exhibit slower convergence, requiring up to 620k iterations of $L_2$ pretraining on cropped images for the model to fully converge.

Evaluation results are presented in Figure 1. We treat models as equal if $p$-value on SbS comparison $> 0.05$ following common practice. More details are in Appendix E.

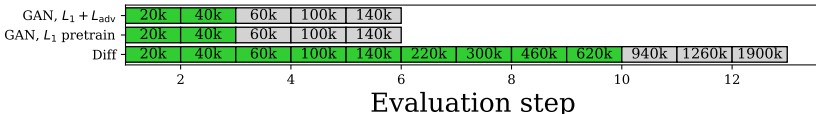

Figure 1: SbS comparison between two subsequent checkpoints showcases faster convergence of GAN models. Green corresponds to statistical improvement on the current step, grey to equality. Three evaluations without improvement in a row indicate convergence.

### 4.3 Performance comparison

**Evaluation** To assess the performance of SR models, we report classic full-reference metrics PSNR, SSIM (Wang et al., 2004), LPIPS Zhang et al. (2018) and recent no-reference CLIP-IQA (Wang et al., 2023a) metric in Table 1. Figure 2 presents several representative visual samples. Additionally, we conduct side-by-side (SbS) comparisons between different models and present user preference results in Table 2 and Table 5. We calculate the number of wins of one model over another plus the number of draws divided by two, which we denote as the *win rate*. The win rates are color-coded: Green if the first model is statistically better than the second, red if the first model is statistically worse, and black otherwise. Details about the statistical significance criteria are provided in Appendix E.

**Results** GAN-based SR model is equal or better than diffusion with respect to the majority of evaluation metrics. Yet both approaches produce visually aesthetic samples with sharp edges, small details, and apparent textures. Our SR models are competitive to the baselines from the literature. According to the user preferences (Table 2) GAN-based model outperforms any of the other models considered. We note that full-reference metrics do not always agree with each other and SbS comparison results.

Table 1: Quantitative comparison between GAN, diffusion SR and current state-of-the-art on $\times 4$ image Super Resolution. Red - best, blue - second best result. Diffusion-based SR approaches are marked with $^\diamond$, GAN-based with $^\star$.

| Metrics | Diff$^\diamond$ (ours) | GAN$^\star$ (ours) | SUPIR$^\diamond$ | RealESRGAN$^\star$ | DiffBIR$^\diamond$ | ResShift$^\diamond$ |
|---|---|---|---|---|---|---|
| PSNR ↑ | 26.655 | 26.006 | 24.270 | 24.435 | 24.809 | 27.830 |
| SSIM ↑ | 0.748 | 0.770 | 0.679 | 0.721 | 0.682 | 0.786 |
| LPIPS ↓ | 0.253 | 0.208 | 0.310 | 0.316 | 0.337 | 0.238 |
| CLIP-IQA ↑ | 0.719 | 0.826 | 0.757 | 0.660 | 0.850 | 0.692 |

Table 2: SbS comparison between GAN-based and diffusion-based SR with each other and current state-of-the-art. The values in the table are win rates of Model 1 over Model 2. Green corresponds to statistical advantage, red to statistical disadvantage, black to statistical indifference between two models.

| Model 1 \ Model 2 | Diff (ours) | GAN (ours) | SUPIR | RealESRGAN | DiffBIR | ResShift |
|---|---|---|---|---|---|---|
| Diff (ours) | x | 9.0 | 37.7 | 85.2 | 51.0 | 46.3 |
| GAN (ours) | 91.0 | x | 62.7 | 94.6 | 69.0 | 86.0 |

### 4.4 Impact of Text Conditioning

We train text-conditioned emodels with the same training setup used for unconditional models. Our experiments cover two text encoders of different nature: our internal CLIP-like model called XL and UMT5 Text-conditioned models have slightly more parameters than unconditional models due to additional transformer blocks. We conduct an SbS comparison between text-conditioned and unconditional models for both paradigms at all stages of training. According to the results in Figure 3 human annotators show preference neither for text-conditional models nor for unconditional.

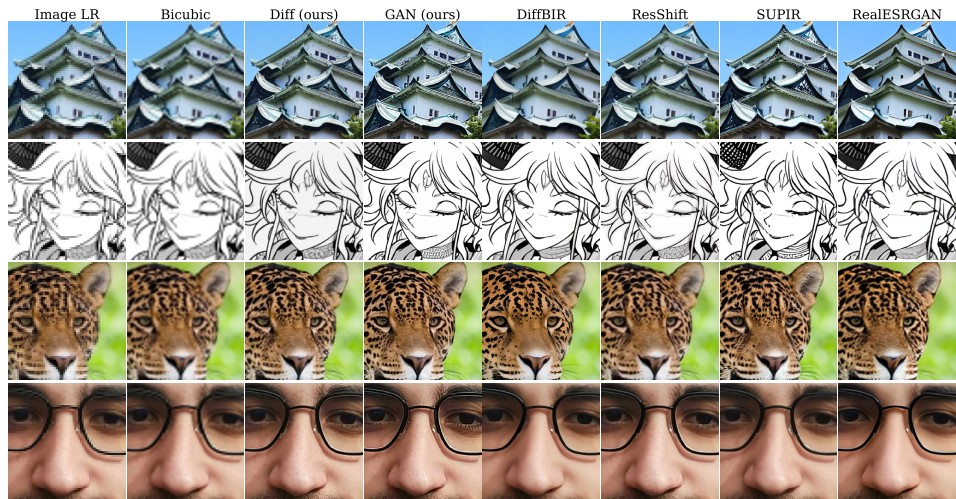

Figure 2: Visual comparison between GAN and diffusion SR model from our work and the baselines on SR($\times 4$). Zoom in for the best view.

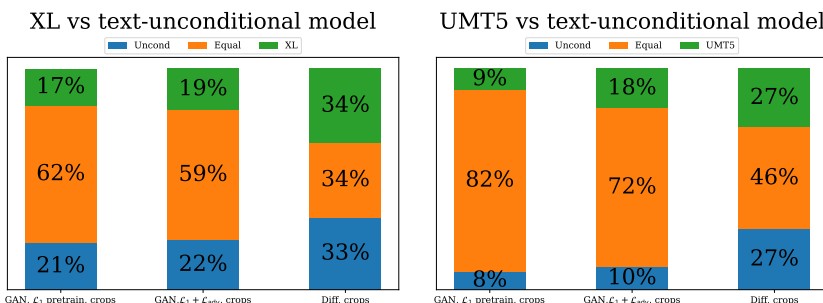

Figure 3: SbS comparison between text-conditional (based on a proprietary model called XL and UMT5) and unconditional SR models. Bar plots show that additional text-conditioning does not noticeably improve perceived image quality

We note that we use semantic image-level captions. Global-level information appears to be not very useful for the SR task. Quality-aware prompts could be beneficial, especially for stronger corruptions (Yu et al. (2024); Yang et al. (2023)). We leave verification of the usefulness of various SR-specific image captioning techniques outside the scope of this work.

### 4.5 ISR FOR A COMPLEX DEGRADATION MODEL AND THE IMPACT OF TRAINING AUGMENTATIONS

Above, all methods were applied to the SR($\times 4$) task, which is essentially an inversion of a fixed downscaling. However, many contemporary works study the task of Blind Super Resolution (BSR), which assumes a more complex degradation model involving multiple, typically unknown corruptions. Real-ESRGAN Wang et al. (2021) is a popular degradation model emulating various degradations that occur in low-quality images. Therefore, following the standard practice, we produce LR inputs via the Real-ESRGAN pipeline.

In our work, we study the impact of augmentations for performance on a simple SR($\times 4$) task and the problem of more general degradation removal. We train both GAN and diffusion models with Real-ESRGAN augmentations in the same way as for the vanilla SR task.

We observe that $L_1$ pretraining and training with the adversarial loss for GAN models converge with the same number of training steps as for training without augmentations (see Figure 4). However, with Real-ESRGAN augmentations, the diffusion SR model requires even more extended training, and its performance saturates only after 940k iterations. The augmentations appear to noticeably slow down the convergence of the diffusion model.

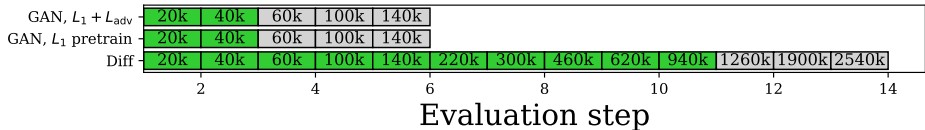

Evaluation step

Figure 4: SbS comparison between two subsequent checkpoints. Green corresponds to statistical improvement, grey to equality. Three evaluations without improvement in a row indicate convergence.

**Results** Pretraining with augmentations is detrimental to the performance on vanilla SR task. The inversion of a non-deterministic, complex degradation process is an inherently more challenging task since the network has to account for multiple degradations of varying magnitude. Therefore, it is natural to expect that models specialized in inverting downscaling without additional degradations better represent sharp edges and fine details. This intuition agrees with the experimental results in Table 3 for SR($\times$4).

At the same time, pretraining with augmentations appears to be necessary for the SR task with additional corruptions. According to the evaluation results in Table 3, pretraining with augmentations improves robustness to corruptions for both GAN and diffusion SR models. However, metrics are still far below the ones for the simple upscaling.

Table 3: Impact of augmentations. Red - best, blue - second best result.

| Dataset | Metrics | Diff (no aug) | Diff (w aug) | GAN (no aug) | GAN (w aug) |
|---------|---------|---------------|--------------|--------------|-------------|
| SR($\times$4) | PSNR | 26.655 | 24.633 | 26.006 | 24.228 |
| | SSIM | 0.748 | 0.674 | 0.770 | 0.761 |
| | LPIPS | 0.253 | 0.330 | 0.208 | 0.294 |
| | CLIP-IQA | 0.719 | 0.773 | 0.706 | 0.751 |
| SR($\times$4) + degradations | PSNR | 22.012 | 22.387 | 22.075 | 22.459 |
| | SSIM | 0.560 | 0.518 | 0.570 | 0.607 |
| | LPIPS | 0.576 | 0.502 | 0.557 | 0.427 |
| | CLIP-IQA | 0.159 | 0.513 | 0.144 | 0.730 |

We show a couple of samples for the dataset with Real-ESRGAN-augmented pipeline in Figure 5. GAN-based and diffusion-based SR models yield samples of similar quality. GAN-based SR from our study appears to be the best in terms of full-reference metrics. At the same time, human annotators show preference for SR models with diffusion priors (DiffBIR, SUPIR). We speculate that presence of rich semantic information is more beneficial in the presence of degradations.

Table 4: Quantitative comparison between GAN, diffusion SR and current state-of-the-art on SR($\times$4) with Real-ESRGAN degradations. Red - best, blue - second best result. Diffusion-based SR approaches are marked with $^\diamond$, GAN-based with $^\star$.

| Metrics | Diff$^\diamond$ (ours) | GAN$^\star$ (ours) | SUPIR$^\diamond$ | RealESRGAN$^\star$ | DiffBIR$^\diamond$ | ResShift$^\diamond$ |
|---------|------------------------|---------------------|------------------|---------------------|---------------------|----------------------|
| PSNR $\uparrow$ | 22.387 | 22.459 | 19.714 | 21.071 | 21.996 | 21.309 |
| SSIM $\uparrow$ | 0.518 | 0.607 | 0.491 | 0.543 | 0.544 | 0.472 |
| LPIPS $\downarrow$ | 0.502 | 0.427 | 0.479 | 0.472 | 0.457 | 0.535 |
| CLIP-IQA $\uparrow$ | 0.513 | 0.730 | 0.747 | 0.656 | 0.781 | 0.585 |

### 4.6 ARTIFACTS

Both SR paradigms suffer from specific artifacts. GAN-based SR models are prone to oversharpening and generating unnatural textures. On the other hand, diffusion-based SR is usually inferior at generating high-frequency details. In addition, diffusion-based SR models tend to generate pale, dim images when compared to the original. We provide visualizations of these common artifacts in Figure 13 and Figure 14.

### 5 ABLATIONS

**Training with Adversarial Loss** In agreement with the literature, we observe that training with adversarial loss is essential for single-step SR to generate sharp details and realistic textures. Human

Table 5: SbS comparison between GAN-based and diffusion-based SR with each other and current state-of-the-art. The values in the table are win rates of Model 1 over Model 2. Green corresponds to statistical advantage, red to statistical disadvantage, black to statistical indifference between two models. Diffusion-based SR approaches are marked with $^\diamond$, GAN-based with $^\star$.

| Model 1 \ Model 2 | Diff$^\diamond$ (ours) | GAN$^\star$ (ours) | SUPIR$^\diamond$ | RealESRGAN$^\star$ | DiffBIR$^\diamond$ | ResShift$^\diamond$ |
|---|---|---|---|---|---|---|
| Diff (ours) | x | 34.0 | 18.4 | 71.7 | 29.3 | 72.1 |
| GAN (ours) | 66.0 | x | 30.0 | 66.2 | 30.6 | 80.2 |

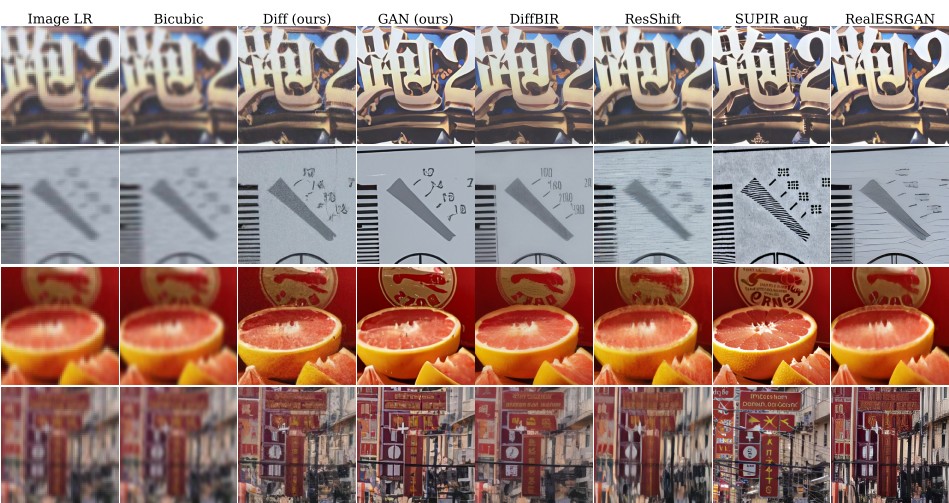

Figure 5: Visual comparison between GAN and diffusion SR model from our work and the baselines on SR($\times$4) + degradations. Zoom in for the best view.

annotators prefer the GAN model when compared to $L_1$ pretrain. According to the results of the SbS comparison a model trained with adversarial loss is preferred in $\sim$95% cases.

**Finetuning on Full-Resolution Inputs** Conventionally, papers on ISR adopt image crops for training since they allow the utilization of larger batches and faster training. However, the inference network accepts full-resolution input rather than crops. Awareness of the overall content could provide additional hints to the network. Therefore, it is questionable whether finetuning on full-resolution inputs may improve quality.

In our experiments, we do not observe any improvement from finetuning on full-resolution images for both SR paradigms. We have tried several choices of learning rate, and for a small enough learning rate, quality remains unchanged according to human preferences according to Table 6, whereas larger learning rates drive both models from good optima and result in degradation of models' performance.

Table 6: Human preferences for GAN SR models without finetuning on full resolution images and with finetuning.

| w/o full resolution finetuning | Equal | w full resolution finetuning |
|---|---|---|
| 4.2% | 90.4% | 5.4% |

**Impact of Perceptual Loss** SRGAN (Ledig et al., 2017a) introduced perceptual loss for GAN training. It is defined as $L_2$ distance between the features of upscaled and the high-resolution image processed via some feature extractor, for instance, VGG19 (Simonyan & Zisserman, 2014). Typically, the addition of perceptual loss to the pixel and adversarial losses is reported to improve the end quality of ISR models. To revisit the impact of perceptual loss, we follow Wang et al. (2021) and use the {conv1, ...conv5} feature maps (with weights {0.1, 0.1, 1, 1, 1}) before activations.

Intriguingly, we observe no significant improvement from using the perceptual loss term. According to the SbS comparison between models trained with and without perceptual loss in Table 7 human annotators do not show preference to model trained with perceptual loss. Based on that, we conclude that perceptual loss has little, if any, impact on the end quality of trained models.

Table 7: Human preferences for GAN SR models trained with and w/o LPIPS loss.

| w LPIPS | Equal | no LPIPS |
|---------|-------|----------|
| 5.0%    | 94.2% | 0.8%     |

## 6 EFFICIENCY

Another important aspect from a practical perspective is the efficiency of the Super Resolution pipeline. While one approach may offer better Super Resolution quality compared to another, this advantage could be outweighed by a much larger inference cost and memory overhead. As one can observe from Table 8, models differ by orders of magnitude in terms of the total number of parameters involved in all stages of the inference workflow. Some approaches can be easily deployed on mobile devices, while others necessitate a compute accelerator with a large enough amount of memory.

We carried out inference measurements on an NVIDIA A100 GPU for upscaling a single image from $256 \rightarrow 1024$ (batch size is 1) and report the latency in Table 8. GAN-based methods produce a high-resolution image given a low-resolution input in a fraction of a second, whereas diffusion-based SR with generative priors requires dozens of seconds.

Table 8: Runtime and number of parameters (during inference) of compared models. Some of the models are consisted of several stages with parameters distributed among them. The benchmarking and parameter-counting protocol are explained in greater detail in Appendix H.5.

|                   | Diff (ours)     | GAN (ours)      | SUPIR            | RealESRGAN        | DiffBIR          | ResShift        |
|-------------------|-----------------|-----------------|------------------|-------------------|------------------|-----------------|
| Runtime, (s)      | $3.20 \pm 0.21$ | $0.24 \pm 0.02$ | $18.11 \pm 0.85$ | $0.065 \pm 0.003$ | $24.69 \pm 0.03$ | $1.19 \pm 0.03$ |
| # Parameters, (M) | 631             | 614             | 17846            | 17                | 1683             | 174             |

## 7 LIMITATIONS

This work considers the standard SR task and SR in the presence of corruptions from a predefined degradation pipeline. The evaluation on the real-world low-quality images, such as the ones from RealLR200 Wu et al. (2023) and RealPhoto60 Yu et al. (2024) datasets is left for future work.

We also do not consider cases when strong generative capabilities and the ability to sample diverse outputs for a given input are required. We hypothesize that in this setup mode collapse can be an issue for GAN-based models. It is also possible that, in these cases, textual information could be necessary for accurate reconstructions. However, we do not consider these facts or the exact turning point when such model capabilities become required.

Another interesting question not covered in our investigation is the study of scaling behavior for both GAN and diffusion SR paradigms with respect to the model size and amount of training data and compute. In preliminary experiments we observed that the training parameters adopted in our main training setup do not transfer immediately to other model sizes and have to be tuned for specific experimental setup.

## 8 CONCLUSION

In our work, we performed a systematic comparison between GAN and diffusion ISR under a fair comparison setup when both approaches are matched in model size and training data available. Our results suggest that GANs can achieve the quality level of modern diffusion models if trained with a similar protocol. At the same time, GAN-based upscalers offer several practical advantages - faster training and single-step inference instead of an iterative denoising procedure.

We hope that our work encourages researchers to conduct more careful examination when introducing new methods. Since the amount of available resources and data is constantly growing, more recent works have the advantage of having more compute and higher-quality data relative to the prior art. Therefore, it is vital to identify whether the improvement comes from one approach's superiority over the other or could be attributed merely to the increase of scale.

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

## A    BACKGROUND ON GENERATIVE PARADIGMS

In this section, we provide a mathematical background for the generative models used in our study.

### A.1    GENERATIVE ADVERSARIAL NETWORKS

To produce samples that resemble a target distribution, one can train an auxiliary critic network, known as a discriminator, to distinguish generated data from real data. The discriminator, denoted as $D_\theta$, is optimized alongside the generator, denoted as $G_\theta$, to solve the following adversarial min-max problem:

$$\min_{\theta_G} \max_{\theta_D} \mathbb{E}_{I^{out} \sim p(I^{out})}[\log D_{\theta_D}(I^{out})] + \mathbb{E}_{I^{in} \sim p(I^{in})}[\log(1 - D_{\theta_D}(G_{\theta_G}(I^{in})))] \tag{5}$$

where $I^{in}$ and $I^{out}$ are the input and output spaces respectively. For instance, $I^{in}$ can be the space of random noise or low-resolution images (for the Super Resolution problem) and $I^{out}$ may correspond to the space of natural images. In the case of successful training, the generator learns to produce samples that are difficult to classify by $D_{\theta_D}$ and resemble the target distribution.

This approach was introduced in Goodfellow et al. (2014) and subsequently became a leading generative paradigm for an extended period. GANs have demonstrated the capability to generate high-quality, photorealistic samples Karras et al. (2017; 2019; 2020). However, they are known to suffer from issues such as unstable training and mode collapse Wiatrak et al. (2020).

### A.2    DIFFUSION MODELS

Diffusion models (Ho et al. (2020)) have emerged as a promising technique in the landscape of generative modeling. These models are primarily inspired by concepts from nonequilibrium thermodynamics, leveraging the idea of diffusion processes to gradually transform simple distributions into complex data distributions and vice versa.

The forward process in diffusion models is a mechanism that systematically transforms structured data into noise through a series of discrete steps. Starting with an original data point $z_0$, this process involves a Markov chain where each step $z_t$ given $z_{t-1}$ is defined by a Gaussian distribution

$$q(z_t|z_{t-1}) = \mathcal{N}(z_t; \sqrt{\alpha_t} z_{t-1}, (1 - \alpha_t)I), \tag{6}$$

with $\alpha_t$ controlling the noise level. As the process progresses, the data is incrementally corrupted, ensuring the distribution of $z_t$ given $z_0$ remains Gaussian,

$$q(z_t|z_0) = \mathcal{N}(z_t; \sqrt{\gamma_t} z_0, (1 - \gamma_t)I), \tag{7}$$

where $\gamma_t = \prod_{i=1}^t \alpha_i$. This results in $z_t$ becoming increasingly dominated by noise, converging to a standard Gaussian distribution as $t$ approaches $T$. The formulation allows for the derivation of the posterior $q(z_{t-1}|z_t, z_0) = \mathcal{N}(z_{t-1}; \mu_t, \sigma_t^2 I)$, providing a basis for the reverse process that aims to denoise the data back to its original form.

The backward diffusion process, or reverse process, in diffusion models, is essential for generating structured data from noise by reversing the forward diffusion steps. Starting from a noise sample $z_T \sim \mathcal{N}(0, I)$, this process iteratively denoises the sample through a sequence of steps $z_T, z_{T-1}, \ldots, z_0$, transforming it back into structured data $z_0$. Each reverse step $p_\theta(z_{t-1}|z_t)$ is modeled as a Gaussian distribution:

$$p_\theta(z_{t-1}|z_t) = \mathcal{N}(z_{t-1}; \mu_\theta(z_t, t), \sigma_\theta(z_t, t)I), \qquad (8)$$

where $\mu_\theta$ and $\sigma_\theta$ are parameterized by a neural network, trained to approximate the reverse of the forward noise addition. The training objective typically maximizes a variational lower bound on the data log-likelihood, encouraging the learned reverse process to closely match the true reverse process. The posterior distribution from the forward process $q(z_{t-1}|z_t, z_0) = \mathcal{N}(z_{t-1}; \mu_t, \sigma_t^2 I)$ guides the reverse process's learning, ensuring that each step effectively reduces the noise incrementally added during the forward process.

This objective can be reduced (Ho et al. (2020)) to a noise prediction one. Given the original data point $z_0$ and its noisy version $z_t$ at step $t$, produced by the forward process $q(z_t|z_0) = \mathcal{N}(z_t; \sqrt{\gamma_t}z_0, (1 - \gamma_t)I)$, the model is tasked with predicting the noise $\epsilon$ that was added. The neural network, parameterized by $\theta$, outputs the predicted noise $\epsilon_\theta(z_t, t)$, aiming to match the actual noise added during the diffusion step. The training objective minimizes the mean squared error between the predicted noise and the true noise:

$$L_{\text{diff}} = \mathbb{E}_{z_0,\epsilon\sim\mathcal{N}(0,I),t\sim\mathcal{U}[0,1]} \left[ \|\epsilon - \epsilon_\theta(z_t, t)\|^2 \right], \qquad (9)$$

where $z_t = \sqrt{\gamma_t}z_0 + \sqrt{1 - \gamma_t}\epsilon$ and $\epsilon \sim \mathcal{N}(0, I)$.

Finally, the diffusion framework can be extended to conditional modeling via appropriate conditioning mechanisms. By incorporating conditional information $c$, such as class labels, text prompts or images, the model is guided to generate data that is contextually relevant and aligned with the specified conditions.

## B QUALITATIVE COMPARISON

Below in Figures 6 and 7, we show additional qualitative comparisons between GAN and SR models from our work. Images are chosen at random from validation datasets.

## C ARCHITECTURE DETAILS

We employ Efficient U-Net architecture introduced in Imagen Saharia et al. (2022) for GAN and diffusion SR models. The number of channels and residual blocks for each resolution is the same as in the $256 \times 256 \rightarrow 1024 \times 1024$ Super Resolution model. The exact model configuration is presented in the listing below.

```
blocks=[
    {
        "channels": 128,
        "strides": (2, 2),
        "kernel_size": (3, 3),
        "num_res_blocks": 2
    },
    {
        "channels": 256,
        "strides": (2, 2),
        "kernel_size": (3, 3),
        "num_res_blocks": 4
    },
    {
        "channels": 512,
        "strides": (2, 2),
        "kernel_size": (3, 3),
        "num_res_blocks": 8
    },
    {
        "channels": 1024,
        "strides": (2, 2),
```

```
23          "kernel_size": (3, 3),
24          "num_res_blocks": 8
25      }
26 ]
```

Listing 1: UNet model configuration

The core difference between GAN and diffusion SR models is the absence of timestep embedding and concatenation of the input noise for the GAN SR model.

For text-conditional models, image caption is first processed via text encoder, and the condition processing module outputs both the tokenized prompt (sequence of embeddings) and the pooled embedding. Text condition is forwarded to the SR model via scale-shift modulation in the residual blocks and cross-attention in transformer blocks. Cross-attention acts only at the lowest resolution in encoder, decoder, and middle blocks. There is no self-attention between image patches.

The model with the smallest number of trainable parameters is a text-unconditional GAN generator with 614M trainable parameters, and the largest one is diffusion U-Net, conditioned on UMT5 embeddings with 696M trainable parameters. Therefore, it is fair to say that the models considered are of the same scale.

For training with adversarial loss, we adopt the discriminator from Real-ESRGAN Wang et al. (2021) with the only difference that the number of channels is multiplied by 2 relative to the network used in the original work. The generator model used in our work is significantly larger than the RRDBNet Wang et al. (2018) from RealESRGAN. Therefore, we have decided to increase the capacity of the discriminator. GAN is trained from scratch and for finetuning on full resolution images we start from the last discriminator checkpoint from training on crops.

## D  TRAINING DETAILS

We use a linear warmup schedule over 1000 training steps to train the diffusion model on image crops, followed by a constant learning rate of 3e-5. For full-resolution inputs, we decrease the learning rate to 1e-6 and increase the duration of the warmup phase to 10k training steps.

For $L_1$ pretrain of GAN SR we use the same learning rate schedule as for diffusion but with a higher learning rate - 2e-4. When training with adversarial loss we adopt the same learning rate 1e-4 both for generator and discriminator. Finetuning on high-resolution images is conducted with a learning rate of 1e-5.

In both cases, models are trained with Adam  Kingma & Ba (2017) optimizer without weight decay with $\beta_1, \beta_2 = (0.90, 0.99)$. For more stable training, we follow common practice and adopt an exponential moving average (EMA) with a factor of 0.999.

Our experiments on crops were performed on two nodes with 8 NVIDIA A100 with 80Gb of VRAM. Distributed communication is performed via Open MPI Message Passing Interface Forum (2021). The optimal batch size for model training is 16, and we use two gradient accumulation steps to have a total batch size of 512.

140k iterations for pretraining and training with the adversarial loss for the GAN-based SR model take approximately three days. Training of diffusion model takes about a month with the same resources.

We adopt Fully Sharded Data Parallel Zhao et al. (2023b) for parameter and optimizer state sharding to reduce memory consumption and allow working with larger batch sizes.

## E  HUMAN EVALUATION

Our main tool for training early stopping is side-by-side (SbS) comparisons of Super Resolution results between two subsequent checkpoints of the same model. For that, we use an internal crowd-sourcing platform with non-expert assessors. Before labeling, each candidate assessor must pass an exam and achieve at least an 80% accuracy rate among a pre-defined set of 20 assignments. After

that, we ask assessors to select one of the two high-resolution images shown Side-by-Side based on the following evaluation criteria placed in order of their importance:

- The presence of color distortions;
- The presence of added artifacts such as exploded pixels and unnatural textures;
- Image blurriness/detalization;
- Color saturation;
- The level of noise.

The assessed high-resolution images are generated from the same low-resolution image. We also provide the initial low-resolution image as a condition to assess potential color distortions. Figure 8 showcases the main labeling interface. Users can also click on any images to open them in full-screen mode and zoom in to better consider the finer details. After that, it becomes possible to toggle between two high-resolution images using hotkeys to assess minor differences in an almost pixel-wise manner.

From a mathematical point of view, human evaluation is a statistical hypothesis test. In particular, we are using a two-sided binomial test and its implementation from `scipy` Virtanen et al. (2020) library to test the null hypothesis of whether the two given models are equal in ISR (accessors follow a special set of instructions when making their decisions). More precisely,

$$p-\text{value} = \text{binom\_test}((n_{\text{wins},1} + n_{\text{eq}}//2, n_{\text{wins},2} + n_{\text{eq}}//2), \ p = 0.5), \tag{10}$$

where $n\_wins_i$ - is the number of generations produced by model $i$ and preferred by accessors, $n\_eq$ - is the number of generations whose quality is indistinguishable according to our guideline. We reject the null hypothesis if $p\_value$ is less than $0.05$, *i.e.*, at the 5% significance level.

## F  DATASET PREPARATION

We start from a proprietary pool of several billion image-text pairs, initially collected to train image-text models. The data pool originally contained samples with various quality images, texts, and image-text relevance. The following filtering procedure aims to select high-quality samples suitable for training competitive Super Resolution models.

First, we select samples with images with exactly 1024 px of height or width and not less than 1024 px in the other direction. For non-square images, we perform a center crop of $1024 \times 1024$ px. This resolution allows us to achieve a reasonable trade-off between data quality and quantity. After that, we pre-calculate a set of scores produced by multiple learned predictors for each image-text pair. These predictors estimate the quality of images, texts, and image-text relevance.

Image quality predictors are fully connected classifiers, trained on top of image and text features, extracted by our proprietary visual foundation model. In this work, we use estimators of image quality trained on PIPAL Gu et al. (2020), KonIQ-10k Hosu et al. (2020) and KADID-10k Lin et al. (2019) datasets, watermark and NSFW content detectors trained on internal data.

We use OpenCLIP Ilharco et al. (2021); Cherti et al. (2023) ViT-G/14 to obtain image-text relevance scores.

We train quality predictor and English language detector for text based on our internal foundation model and data.

After that, we filter the data based on the predictors mentioned above so that the resulting dataset size does not contain obviously bad samples according to the predictor models. We also aim for the resulting dataset size sufficient to train any of our models for up to several dozen epochs. The resulting dataset consists of 17 million $1024 \times 1024$ px images and corresponding captions.

## G  SUPER RESOLUTION OF SYNTHETIC IMAGES

To test the robustness of Super Resolution models on the out-of-distribution (OOD) data, we apply both models for upscaling of $4\times$ downscaled 1024px generations produced by YaART Kastryulin

et al. (2024) and SDXL Podell et al. (2024) models as two examples of modern text-conditional generative models. The former is a cascaded pixel diffusion model. The latter operates in latent space, and the generated sample is transformed to pixel space with the help of a decoder network.

We generate samples using prompts from DrawBench Saharia et al. (2022) and YaBasket Kastryulin et al. (2024) with both models. (Add details about the generation setup - sampler, number of steps, condition scale to Appendix). LR images are obtained by downscaling 1024px generations to 256px via `cv2.INTER_AREA`.Even though both models produce photorealistic samples of high quality, generated images could still be different from the data observed during training. The discrepancy between training and evaluation data may negatively affect the performance of SR models and cause artifacts.

**Results**    Below, we provide only qualitative evaluation since there is no actual ground truth for this setting, as generative models produce outputs that only approximate real-world images. Hence, the calculation of no-reference and full-reference metrics is not very meaningful. Representative examples are shown in Figures 9,10, 11, 12. Models trained without augmentations appeared to be robust enough for evaluation on synthetic images while providing sharper edges and better representing high-frequency details.

## H    ADDITIONAL DETAILS

### H.1    SAMPLER HYPERPARAMETERS SETTING FOR DIFFUSION MODEL

In our experiments, we evaluate the diffusion model using DPM-Solver++(2M) Lu et al. (2023) with 13 sampling steps. This setup demonstrated robust outcomes with a relatively modest number of steps, effectively balancing speed and output quality. To ensure this, we evaluated our diffusion model with different samplers (DDIM Song et al. (2020), UNIPC Zhao et al. (2023a)) and a varying number of sampling steps. The results are presented in Table 9. It is evident from the results that increasing the number of steps beyond those used in our DPM-Solver++ baseline (13 steps) does not enhance performance, whereas reducing the number of steps compromises visual quality.

Table 9: Comparison of different sampler settings to the one used in the experiments (DPM-Solver++, 13). Side-by-side results and basic metrics are introduced. The setting used in this work is optimal in terms of speed and output quality.

| Sampler, #steps | Wins | Loses | Ties | p-value | PSNR ↑ | SSIM ↑ | LPIPS ↓ | CLIP-IQA ↑ |
|---|---|---|---|---|---|---|---|---|
| DPM-Solver++, 13 | - | - | - | - | 26.66 | 0.748 | 0.253 | 0.719 |
| DPM-Solver++, 6 | 0.192 | 0.479 | 0.329 | 0.0 | 27.77 | 0.793 | 0.229 | 0.666 |
| DPM-Solver++, 64 | 0.287 | 0.283 | 0.430 | 1.0 | 26.28 | 0.730 | 0.275 | 0.737 |
| UniPC, 6 | 0.138 | 0.354 | 0.508 | 0.001 | 27.58 | 0.786 | 0.232 | 0.657 |
| UniPC, 13 | 0.367 | 0.338 | 0.295 | 0.70 | 26.50 | 0.740 | 0.259 | 0.728 |
| UniPC, 32 | 0.267 | 0.254 | 0.479 | 0.85 | 26.28 | 0.730 | 0.274 | 0.740 |
| UniPC, 64 | 0.425 | 0.425 | 0.150 | 1.0 | 26.26 | 0.730 | 0.277 | 0.742 |
| DDIM, 100 | 0.358 | 0.354 | 0.288 | 1.0 | 26.60 | 0.745 | 0.256 | 0.730 |
| DDIM, 500 | 0.417 | 0.400 | 0.183 | 0.85 | 26.35 | 0.731 | 0.269 | 0.741 |
| DDIM, 1000 | 0.421 | 0.383 | 0.183 | 0.61 | 26.32 | 0.729 | 0.272 | 0.735 |

### H.2    FINE-TUNING ON FULL-RESOLUTION INPUTS

Our experiments show that fine-tuning on full-resolution inputs brings no benefit compared to training only on crops. Here, we present metrics computed based on the best checkpoints trained on crops only and full-resolution images (Table 10), as well as a side-by-side comparison (Table 11). This can be explained by the fact that both full-resolution and cropped inputs contain the same low-level information, and the global-level semantic information appears to be irrelevant for the SR task.

We experimented with different learning rates for the full-resolution fine-tune: 1e-4, 1e-5 for GAN, and 3e-5, 1e-6 for diffusion. While training models with the rates used for training on crops (1e-4 for GAN and 3e-5 for diffusion) degrades generation quality (color-shifting and blurred generation), fine-tuning both models even with the smallest rates for 300k iterations leads to no statistical

Table 10: Comparison of only-crops traing vs fine-tuning on full-resolution images based on basic metrics.

| Models | PSNR ↑ | SSIM ↑ | LPIPS ↓ | CLIP-IQA ↑ |
|---|---|---|---|---|
| GAN (crops) | 26.00 | 0.770 | 0.208 | 0.806 |
| GAN (full-res) | 28.27 | 0.811 | 0.182 | 0.790 |
| Diffusion (crops) | 26.66 | 0.748 | 0.253 | 0.719 |
| Diffusion (full-res) | 26.83 | 0.761 | 0.236 | 0.715 |

Table 11: Side-by-side comparison of only-crops training vs fine-tuning on full-resolution images.

| | Wins | Loses | Ties | p-value |
|---|---|---|---|---|
| GAN (crops) vs GAN (full-res) | 0.425 | 0.441 | 0.133 | 0.847 |
| Diff (crops) vs Diff (full-res) | 0.213 | 0.320 | 0.467 | 0.109 |

improvements. We provide the results of human evaluations against crop-trained baselines for all mentioned setups in Table 12.

Table 12: Side-by-side comparison of only-crops training vs fine-tuning on full-resolution images with different learning rates.

| | Wins | Loses | Ties | p-value |
|---|---|---|---|---|
| GAN (crops) vs GAN (full-res, 1e-5) | 0.425 | 0.441 | 0.133 | 0.847 |
| GAN (crops) vs GAN (full-res, 1e-4) | 0.658 | 0.175 | 0.167 | 0.000 |
| Diff (crops) vs Diff (full-res, 1e-6) | 0.213 | 0.320 | 0.467 | 0.109 |
| Diff (crops) vs Diff (full-res, 3e-5) | 0.212 | 0.155 | 0.633 | 0.410 |

### H.3 EARLY STOPPING WITH SIDE-BY-SIDE COMPARISON

In our experiments, we adopted a side-by-side comparison to determine the optimal time for early stopping, when further changes in the output of the SR model become imperceptible to human observers (see Appendix E for more detail). However, automated full-reference metrics are more common in the SR literature, so it is interesting to track their evolution during training. We present the measurement of some full-reference metrics in Table 13. One can observe that during pretraining, PSNR and SSIM continue to slightly improve, whereas for GAN training, the metrics stabilize around a specific value after a certain number of iterations. This indicates that above a certain threshold, further improvement in PSNR/SSIM is hardly discernible.

### H.4 SUPER RESOLUTION ARTIFACTS

We visualize artifacts typical for GAN-based SR and diffusion-based SR in Figure 13 and 14.

### H.5 EFFICIENCY CONSIDERATIONS

**Model Size** Different SR approaches may vary drastically in terms of the inference pipeline structure and the sizes of the models involved. The superior quality of one method over another could be outweighed in real-world applications by the amount of memory needed to run inference. When considering the size of SR models with diffusion priors (SUPIR, DiffBIR), we include the size of the UNet, text encoders, and VAE decoder.

- **Diff** (631 M) = **Denoiser** (631 M)

Table 13: Evolution throughout training of PSNR and SSIM.

| Evaluation Step | L1-pretrain | | GAN | | Diff | |
|---|---|---|---|---|---|---|
| | PNSR ↑ | SSIM ↑ | PNSR ↑ | SSIM ↑ | PNSR ↑ | SSIM ↑ |
| 20k | 28.21 | 0.834 | 25.84 | 0.767 | 20.57 | 0.659 |
| 40k | 28.48 | 0.840 | 26.01 | 0.770 | 20.89 | 0.656 |
| 60k | 28.66 | 0.843 | 26.01 | 0.770 | 21.33 | 0.663 |
| 100k | 28.92 | 0.846 | 26.00 | 0.770 | 22.25 | 0.675 |
| 140k | 29.03 | 0.847 | 26.00 | 0.770 | 22.89 | 0.681 |
| 220k | - | - | - | - | 23.92 | 0.701 |
| 300k | - | - | - | - | 25.66 | 0.733 |
| 460k | - | - | - | - | 26.03 | 0.741 |
| 620k | - | - | - | - | 26.66 | 0.748 |

- **GAN** (631 M) = **Generator** (614 M) + **Discriminator** (17 M)

- **SUPIR** (17846 M) = **SDXL** (3469 M) + **ControlNet** (1332 M) + **LLaVA** (13045 M)

- **RealESRGAN** (21 M) = **Generator** (17 M) + **Discriminator** (4 M)

- **DiffBIR** (1683 M) = **SDv2.1** (1303 M) + **Restoration Module** (17 M) + **ControlNet** (363 M)

- **ResShift** (174 M) = **Denoiser** (119 M) + **AutoEncoder** (55 M)

One can see that different approaches may differ by orders of magnitude in model scale.

**Evaluation Settings**  We conducted all inference measurements on 1 NVIDIA A100 GPU with 80GB of VRAM, batch size 1, PyTorch 2.4.0 Paszke et al. (2019), and CUDA 12.4. We evaluated performance on a 256px → 1024px image super-resolution task.

As mentioned earlier, for our diffusion model, we used 13 steps to generate upscales, while for other diffusion baselines we adopted the default settings from the corresponding papers and repositories: 4 sampling steps for ResShift, 100 sampling steps for SUPIR, and 50 sampling steps for DiffBIR. Note that for these baselines, the reported time spent to produce one image also includes inference of some extra networks such as VAE, LLaVa, BSRNet, etc.

Both our GAN and RealESRGAN models require 1 network evaluation to generate final images.

## I  DATA-SCALING EXPERIMENTS

In the main part we focused on performace comparison in case of data abundance of data. In this section we explore impact of data size on performance of Super Resolution models.

Specifically, we selected subsets of {18k, 180k, 1.8M} samples from the original large training dataset and trained both GAN-based and diffusion-based models for an identical number of steps. We then calculated image quality assessment (IQA) metrics and performed side-by-side comparisons between models trained on the full dataset and those trained on the smaller subsets. Furthermore, we compared the corresponding GANs and diffusion models.

Results of IQA metric evaluation for different amount of training data are reported in Table 14 and Table 15. Additionally, we report results of user preference study between GAN-based and diffusion-based SR models in Tables 16, 17, 18.

Remarkably, the performance of GAN-based SR saturates quickly with respect to the amount of training data. We observed no significant difference in terms of standard SR metrics (PSNR/S-SIM/LPIPS) between the model trained on the smallest amount of data (i.e 18k samples) vs the one trained on 18M images. Only CLIP-IQA shows small improvement. However, Side-by-Side comparison between different GAN-base SR models shows equality with respect to human assessment for models trained on various amounts of data.

Table 14: Quantitative comparison between GAN-based models trained with different amounts of data.

| MetricsDataset size | $1.8 \cdot 10^4$ | $1.8 \cdot 10^5$ | $1.8 \cdot 10^6$ | $1.8 \cdot 10^7$ |
|---|---|---|---|---|
| PSNR ↑ | 25.538 | 26.366 | 25.811 | 26.010 |
| SSIM ↑ | 0.747 | 0.770 | 0.763 | 0.770 |
| LPIPS ↓ | 0.241 | 0.233 | 0.237 | 0.208 |
| CLIP-IQA ↑ | 0.749 | 0.758 | 0.769 | 0.826 |

Table 15: Quantitative comparison between diffusion-based models trained with different amounts of data.

| Dataset size / Metrics | $1.8 \cdot 10^4$ | $1.8 \cdot 10^5$ | $1.8 \cdot 10^6$ | $1.8 \cdot 10^7$ |
|---|---|---|---|---|
| PSNR ↑ | 24.830 | 25.330 | 25.020 | 26.670 |
| SSIM ↑ | 0.665 | 0.697 | 0.689 | 0.748 |
| LPIPS ↓ | 0.351 | 0.328 | 0.330 | 0.253 |
| CLIP-IQA ↑ | 0.620 | 0.602 | 0.647 | 0.719 |

At the same time, the diffusion model appears to be more sensitive to data. Whereas, models trained on 180k, 1.8M match the model trained on large dataset in terms of quality, the one trained on the smallest number of samples appears to be statistically worse. This finding suggests that diffusion-based SR is likely to be more "data-hungry" compared to GAN-based SR.

Finally, our findings indicate that GAN-based models outperform diffusion-based models, regardless of the amount of training data.

## J    LSDIR-TRAINED MODELS

To provide clearer insights into the advantages of GANs over diffusion models in super-resolution (SR), we trained both paradigms using a random sample of 18,000 images from the open-source LSDIR dataset Li et al. (2023). We then calculated image quality assessment (IQA) metrics and performed a side-by-side comparison of the final models. Details can be found in Tables 19 and 20.

Experimental results suggest that the GAN paradigm outperforms diffusion in SR tasks, even when models are trained on different datasets.

We further investigated the influence of the dataset on upscaling quality by conducting side-by-side comparisons between models trained on the LSDIR dataset and those trained on different fractions of our dataset. Results are presented in Tables 21 and 22 below.

For GANs, regardless of the fraction of our dataset used, the resulting model performed better than the one trained on the LSDIR dataset. Conversely, the diffusion model trained on an 18k sample of our dataset performed on par with the LSDIR-trained diffusion model, while the model trained on the full dataset surpassed the LSDIR-trained one.

Table 16: SbS comparison between GAN-based SR models for different sizes of training dataset. The values in the table are win rates of Model 1 over Model 2. Green corresponds to statistical advantage, red to statistical disadvantage, black to statistical indifference between two models.

| Model 2 / Model 1 | $GAN_{1.8 \cdot 10^4}$ | $GAN_{1.8 \cdot 10^5}$ | $GAN_{1.8 \cdot 10^6}$ | $GAN_{1.8 \cdot 10^7}$ |
|---|---|---|---|---|
| $GAN_{1.8 \cdot 10^7}$ | 50.1 | 53.3 | 48.5 | x |

Table 17: SbS comparison between diffusion-based SR models for different sizes of training dataset. The values in the table are win rates of Model 1 over Model 2. Green corresponds to statistical advantage, red to statistical disadvantage, black to statistical indifference between two models.

| Model 1 \ Model 2 | $\text{Diff}_{1.8\cdot10^4}$ | $\text{Diff}_{1.8\cdot10^5}$ | $\text{Diff}_{1.8\cdot10^6}$ | $\text{Diff}_{1.8\cdot10^7}$ |
|---|---|---|---|---|
| $\text{Diff}_{1.8\cdot10^7}$ | 71.4 | 56.1 | 56.3 | x |

Table 18: SbS comparison between diffusion-based SR and GAN-based SR for different sizes of training dataset. The values in the table are win rates of GAN over diffusion. Green corresponds to statistical advantage, red to statistical disadvantage, black to statistical indifference between two models.

| | $1.8\cdot10^4$ | $1.8\cdot10^5$ | $1.8\cdot10^6$ | $1.8\cdot10^7$ |
|---|---|---|---|---|
| GAN \ Diffusion | 90.5 | 86.7 | 88.4 | 91.0 |

Table 19: SbS comparison between diffusion-based SR and GAN-based SR trained on 18k sample of LSDIR dataset. The values in the table are win rates of GAN over diffusion. Green corresponds to statistical advantage, red to statistical disadvantage, black to statistical indifference between two models.

| Model 1 \ Model 2 | Diff-LSDIR |
|---|---|
| GAN-LSDIR | 76.2 |

Table 20: Quantitative comparison between GAN-based and diffusion-based models trained on random 18k sample of LSDIR.

| Metrics \ Model | GAN-LSDIR | Diff-LSDIR |
|---|---|---|
| PSNR ↑ | 24.290 | 26.431 |
| SSIM ↑ | 0.646 | 0.772 |
| LPIPS ↓ | 0.362 | 0.232 |
| CLIP-IQA ↑ | 0.651 | 0.757 |

Table 21: SbS comparison between GAN-based SR models train on LSDIR and different sizes of our training dataset. The values in the table are win rates of Model 1 over Model 2. Green corresponds to statistical advantage, red to statistical disadvantage, black to statistical indifference between two models.

| Model 1 \ Model 2 | $\text{GAN}_{1.8\cdot10^4}$ | $\text{GAN}_{1.8\cdot10^7}$ |
|---|---|---|
| GAN-LSDIR | 35.2 | 28.8 |

Table 22: SbS comparison between diffusion-based SR models train on LSDIR and different sizes of our training dataset. The values in the table are win rates of Model 1 over Model 2. Green corresponds to statistical advantage, red to statistical disadvantage, black to statistical indifference between two models.

| Model 1 \ Model 2 | $\text{Diff}_{1.8\cdot10^4}$ | $\text{Diff}_{1.8\cdot10^7}$ |
|---|---|---|
| Diff-LSDIR | 51.1 | 42.5 |

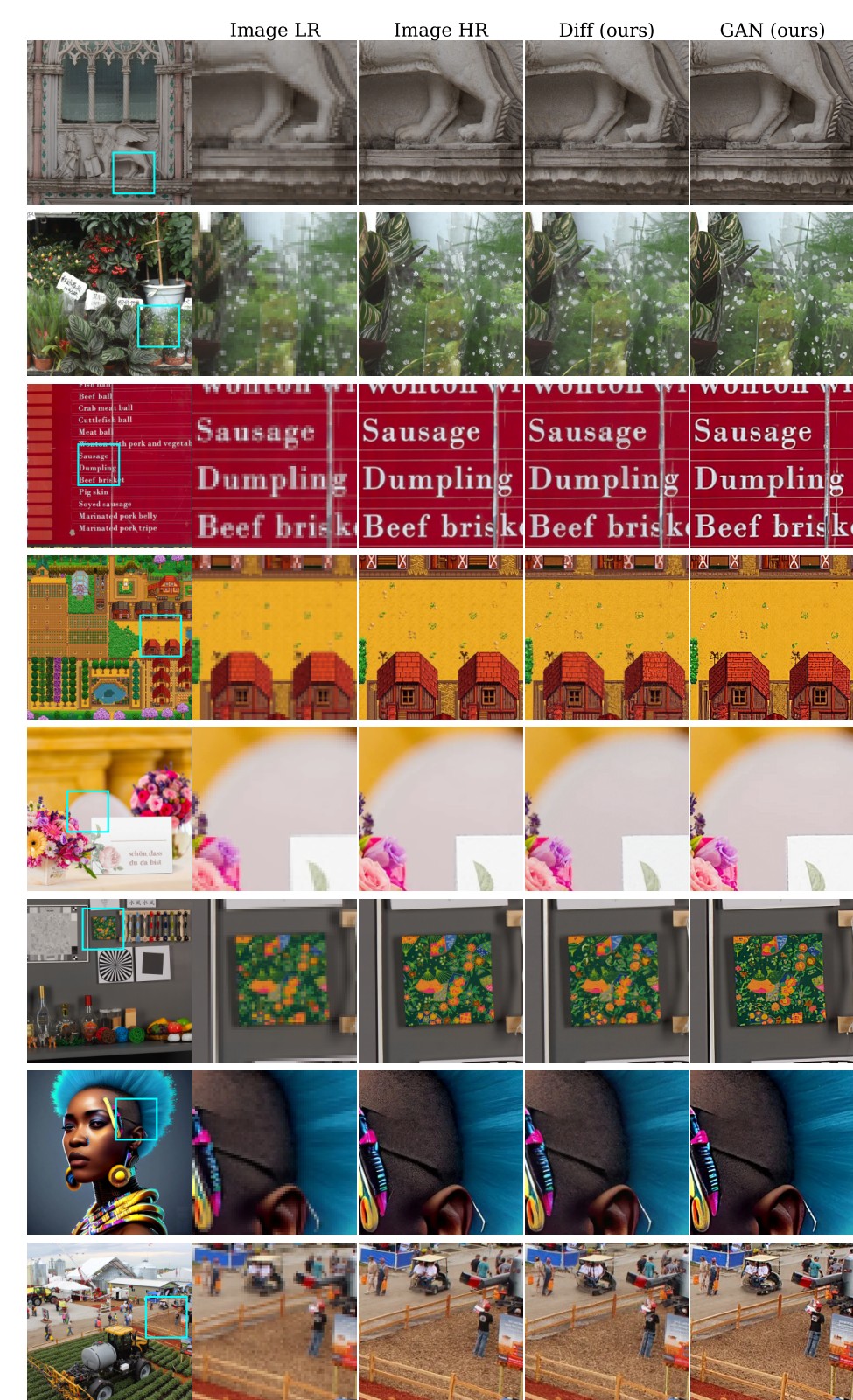

Figure 6: Qualitative comparison between GAN and SR models from our work on the original dataset. Zoom in for the best view.

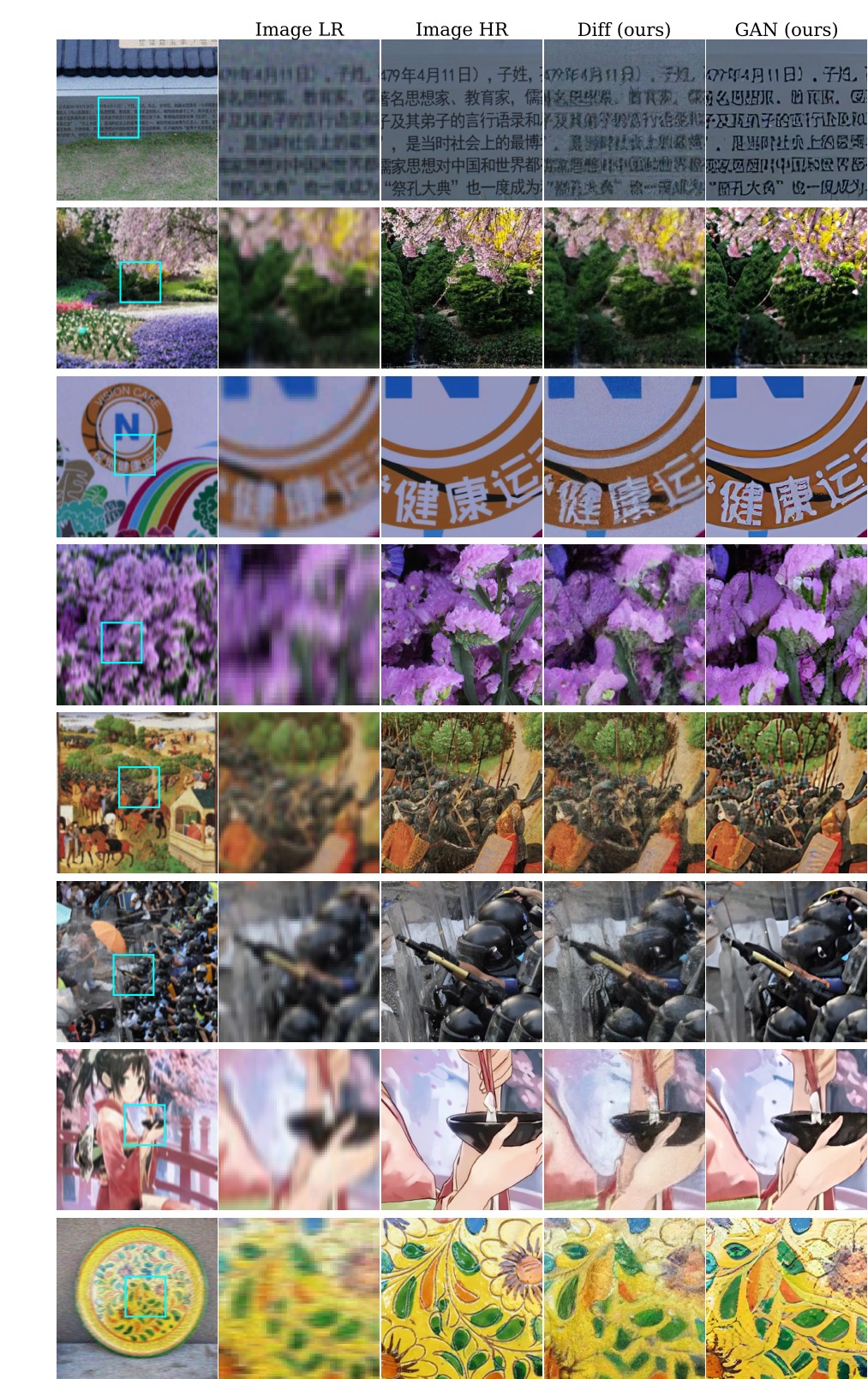

Figure 7: Qualitative comparison between GAN and SR models from our work on a test set with Real-ESRGAN degradations. Zoom in for the best view.

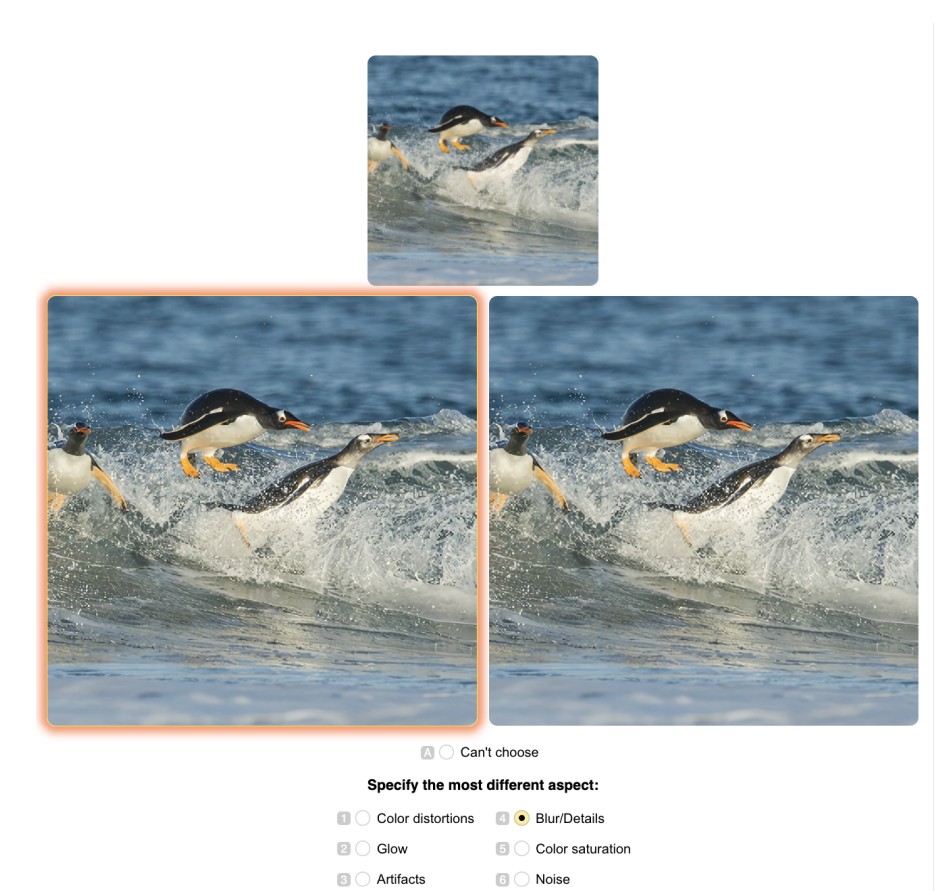

Figure 8: An example of a user interface for Human Evaluation with Side-by-Side comparisons.

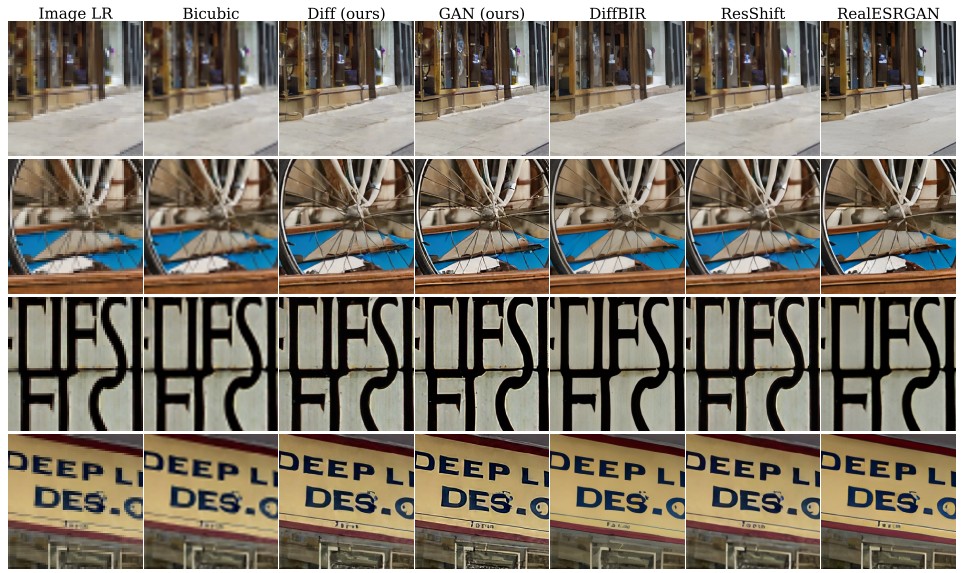

Figure 9: Qualitative comparison between SR models on images generated with SDXL on Draw-Bench prompts. Zoom in for the best view.

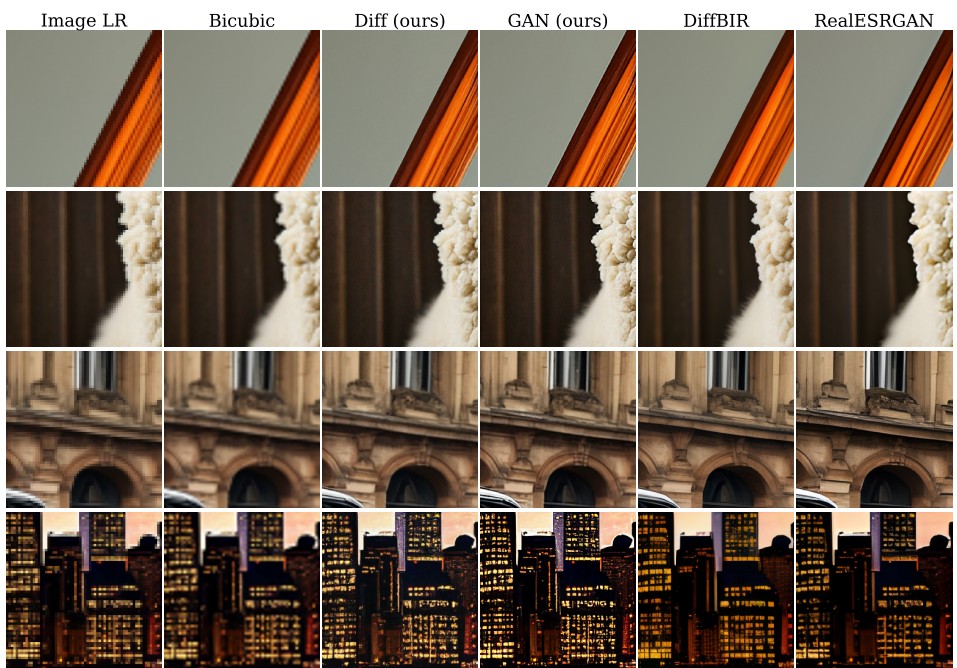

Figure 10: Qualitative comparison between SR models on images generated with YaART on Draw-Bench prompts. Zoom in for the best view.

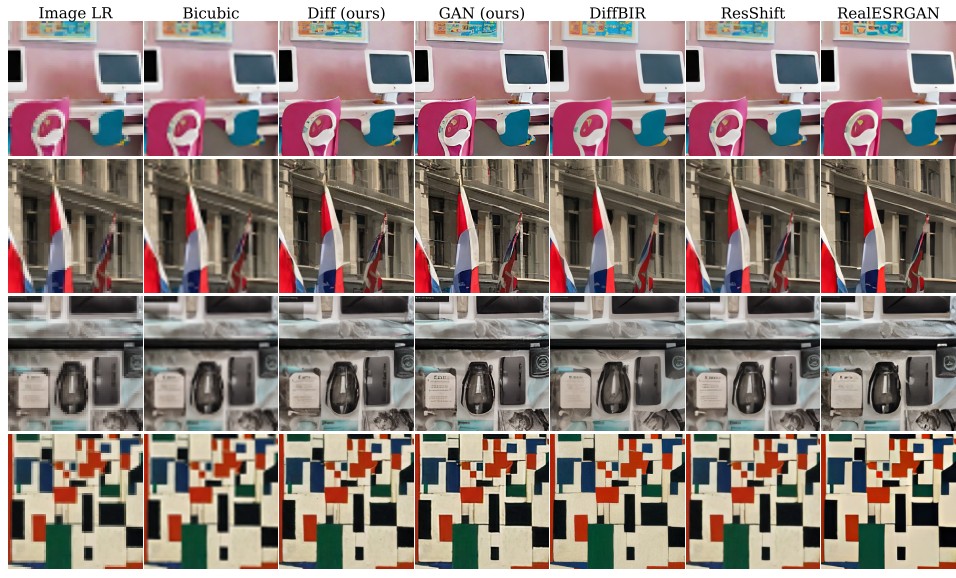

Figure 11: Qualitative comparison between SR models on images generated with SDXL on YaBasket prompts. Zoom in for the best view.

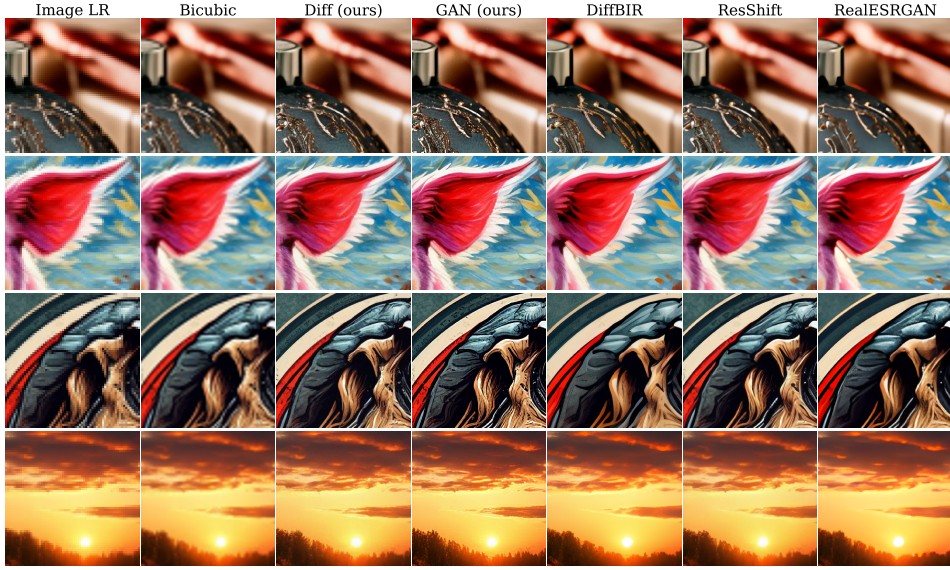

Figure 12: Qualitative comparison between SR models on images generated with YaART on YaBasket prompts. Zoom in for the best view.

Image HR          Diff          GAN

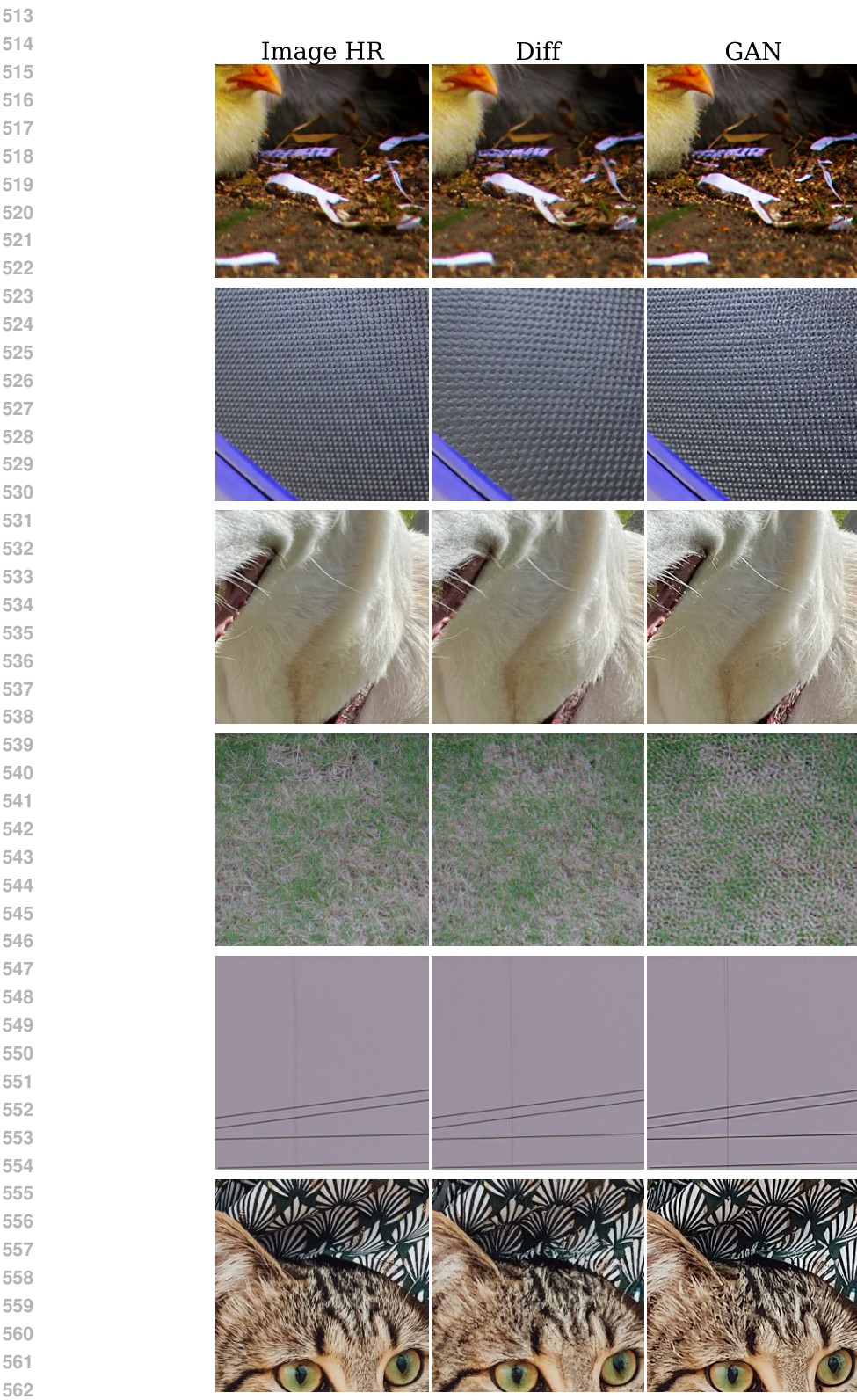

Figure 13: GAN's oversharpening leads to unnatural textures (e.g. fur or grass) and artifacts.

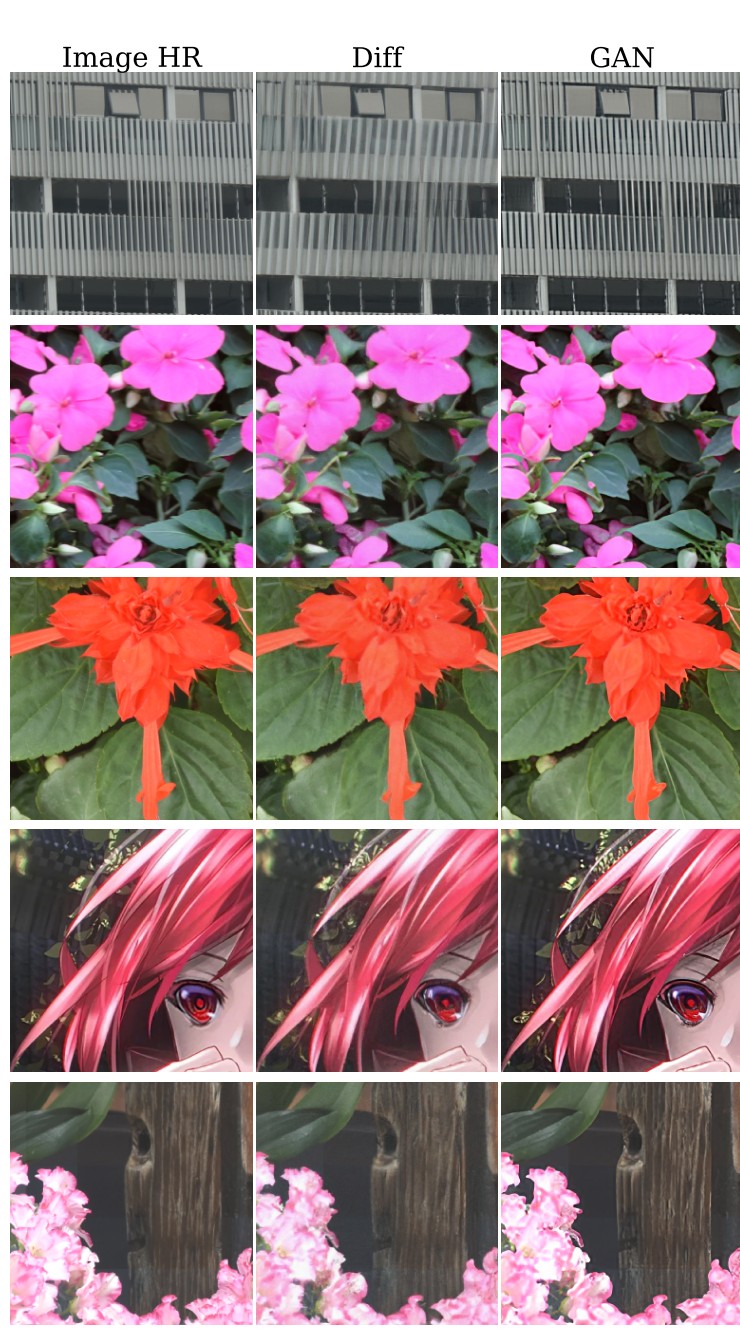

Figure 14: Diffusion is usually worse at high-frequency details (1 row). Additionally, even after big number of training iterations may generate pale, dim images (rows 2-5).

