# OpenReview forum: "Does Diffusion Beat GAN in Image Super Resolution?"
_ICLR.cc/2025/Conference — Submitted to ICLR 2025_

### Official Review · Reviewer_PCgZ · 2024-10-24

**Soundness:** 3
**Presentation:** 2
**Contribution:** 3
**Rating:** 6
**Confidence:** 3

**Summary:**

This paper explores the challenge of conducting a fair comparison between GAN and diffusion models in image super-resolution. It notes that diffusion models often employ larger networks and longer training times than GANs, raising questions about whether their performance is due to inherent advantages or increased resources. Through controlled comparisons, matching architecture, dataset size, and computational budget, the study observes that GANs can achieve results comparable or superior to diffusion models. It also examines factors like text conditioning and data augmentation, finding that commonly used image captions do not significantly impact performance.

**Strengths:**

1. The authors provide valuable insight into the current research trend, highlighting the need for fair comparisons between diffusion and GAN models.
2. They conduct several experiments, considering computational budget, dataset size, and notably, text conditioning as input prompts.
3. The authors incorporate human perception to regulate training, ensuring a fair comparison by controlling the training duration.
4. The experiments are sound, with the authors aiming to maintain consistent conditions, such as using DPM-Solver++ to control the inference steps.

**Weaknesses:**

1. The presentation of this paper could be enhanced with more figures to better illustrate the concepts and experiments, especially those related to text-conditioning.
2. The comparison of super-resolved images in the manuscript and supplementary materials lacks numerical metrics, making evaluation challenging.
3. The current conclusion is somewhat vague. While the authors conduct various ablations between GAN and diffusion, it remains unclear under which specific conditions GAN can truly outperform diffusion, thus it needs to be clarified and streamlined.

**Questions:**

1. The authors use Imagen as the architecture for both GAN and diffusion models in their SR experiments. I question whether it is appropriate to use Imagen for the GAN model, given it wasn't designed for GAN-based SR.
2. My major concern is that the experiments aren't entirely equivalent. Although the authors attempt to balance factors, the GAN model requiring an additional discriminator compared to the diffusion model, the training stability and results heavily depend on this discriminator. It's challenging to ensure true equality between GAN-based methods and diffusion models.

---

> ### Author Response · Authors · 2024-11-25
>
> We are grateful for the reviewer's insightful feedback. We address their concerns and questions below.
>
> **Weaknesses**
>
> > The presentation of this paper could be enhanced with more figures to better illustrate the concepts and experiments, especially those related to text-conditioning.
> In both Diffusion and GAN models, text conditioning is implemented identically through Cross-Attention between image and text tokens together with scale-shift modulation in Residual Convolutional Blocks. This occurs at the lowest resolution within the UNet encoder, decoder, and middle block, as detailed in the "Text Conditioning" section of Section 3. This design choice is standard and consistent with the approaches used in the SR3 [5]  and Imagen [3].
>
> > The comparison of super-resolved images in the manuscript and supplementary materials lacks numerical metrics, making evaluation challenging.
>
> We compare super-resolved images to the high-resolution references via standard IQA metrics and conduct side-by-side comparison between different models  in user preference study as discussed in Section 4.3 paragraph Evaluation. We are open to suggestions about improving the presentation of numerical metrics.
>
> > The current conclusion is somewhat vague. While the authors conduct various ablations between GAN and diffusion, it remains unclear under which specific conditions GAN can truly outperform diffusion, thus it needs to be clarified and streamlined.
>
> The overall conclusion of our study is that GAN-based models are preferable in most cases as they are faster to train and inference while being of same or better quality. Both paradigms have specific artifacts, as discussed in Section 4.6. Depending on the application one type of artifacts can be considered as more severe compared to another.
>
> **Questions**
>
> > The authors use Imagen as the architecture for both GAN and diffusion models in their SR experiments. I question whether it is appropriate to use Imagen for the GAN model, given it wasn't designed for GAN-based SR.
>
> Imagen uses a UNet architecture that generates output with the same dimensions as the input. The key difference between Diffusion-based and GAN-based super-resolution lies in their input and target specifications: Diffusion-based models take a noisy image as input and predict the noise, whereas GAN-based models use a bicubic-upscaled low-resolution image as input, with the high-resolution image as the target. It's important to note that pre-upsampling is a standard approach in super-resolution literature [1, 2], and the UNet architecture has been implemented in GAN-based super-resolution as discussed in [3].
>
> > My major concern is that the experiments aren't entirely equivalent. Although the authors attempt to balance factors, the GAN model requiring an additional discriminator compared to the diffusion model, the training stability and results heavily depend on this discriminator. It's challenging to ensure true equality between GAN-based methods and diffusion models.
>
> We acknowledge the reviewer's concern that the presence of a discriminator in GAN-based super-resolution can complicate comparisons. However, the discriminator is crucial for generating sharp details and ensuring images lie on the manifold of natural images. We utilized the discriminator from Real-ESRGAN and found it effective without requiring extensive tuning. Overall, our study aims to compare the performance of Diffusion-based and GAN-based super-resolution in the most common setting.
>
> References
> ---
>
> [1] Dong, Chao, et al. "Image super-resolution using deep convolutional networks." IEEE transactions on pattern analysis and machine intelligence 38.2 (2015): 295-307.
>
> [2] Kim, Jiwon, Jung Kwon Lee, and Kyoung Mu Lee. "Deeply-recursive convolutional network for image super-resolution." Proceedings of the IEEE conference on computer vision and pattern recognition. 2016.
>
> [3] Wang, Yin. "Single image super-resolution with u-net generative adversarial networks." 2021 IEEE 4th Advanced Information Management, Communicates, Electronic and Automation Control Conference (IMCEC). Vol. 4. IEEE, 2021.

---

> > ### Comment · Reviewer_PCgZ · 2024-11-26
> > **Response to authors**
> >
> > After reviewing the authors' responses, I have no remaining concerns regarding my previous questions. I find this paper to be well-conducted, with clear motivation, and it holds value for the community, particularly for image restoration and synthesis. However, the layout and figures could be improved to enhance understanding for novice readers. Therefore, I will maintain my rating of borderline accept.

---

### Official Review · Reviewer_cAL4 · 2024-10-26

**Soundness:** 3
**Presentation:** 3
**Contribution:** 2
**Rating:** 5
**Confidence:** 4

**Summary:**

This paper investigates whether diffusion-based models truly outperform GAN-based models for the task of Image Super Resolution (ISR) in a fair setup. The authors conduct a rigorous comparison between diffusion and GAN-based models under controlled experimental settings, ensuring that both model types have equivalent architectures, dataset sizes, and computational budgets. In contrast with common belief, the primary findings reveal that GAN-based models can match or even surpass diffusion models in terms of quality when appropriately scaled. The paper also finds that text conditioning has little effect on model performance, while augmentations can sometimes hurt the training of diffusion models.

**Strengths:**

1. The main contribution of this paper is a fair comparison between GAN and diffusion-based ISR models, controlling for architecture, dataset size, and computational resources.

2. The authors perform detailed ablation studies, particularly focusing on the effects of pretraining, augmentations, and training with full-resolution images.

3. The paper explores not only the overall performance of the models but also the impact of various design choices such as text conditioning and augmentation, which can be helpful for future work in the field.

**Weaknesses:**

1. The contribution is a bit limited as there is no really new and impactful insights presented in this work. Additionally, I'm not sure ICLR is a suitable venue for submitting this work, because it lean toward to more empirical side. May be MLSys is a better venue ? Again, I'm not sure.

2. Even though the experiment setting is quite fair, GAN-based models actually have one additional pretraining stage whereas diffusion model has to be trained from scratch, which could be a reason why diffusion-based model lacks behind GAN-based ones.

3. The authors do not report how the GAN-based model perform without L1 pretraining stage, both qualitatively and quantitatively. Also, claiming that training GAN-based models for ISR does not face instability issues is quite a bold claim because without pretraining on L1 loss, I can imagine that it could be really unstable.

4. Though the paper highlights that both GANs and diffusion models benefit from scaling, it does not investigate how these models scale in terms of data. Like, how much data that GAN-based model starts to outperform diffusion ones. This could help to strengthen the work.

**Questions:**

1. Is it possible to do some "pretraining" for diffusion-based models similar to the pretraining on L1 for Gan-based method ? So that it can be more fair in evaluation.

2. As I mentioned above in Weaknesses section (bullet point 4), I would like to see how data scaling affect the performance of both GAN-based and diffusion-based ISR models.

3. I know it's a bit infeasible but showing comparison on other restoration tasks like image dehazing, deblurring, .. to see if the same phenonmena happens can be really valuable and strengthen the work.

---

> ### Author Response · Authors · 2024-11-25
>
> We thank the reviewer for their feedback and provide responses to their concerns hereafter.
>
> **Weaknesses (part1)**
>
> > The contribution is a bit limited as there is no really new and impactful insights presented in this work. Additionally, I'm not sure ICLR is a suitable venue for submitting this work, because it lean toward to more empirical side. May be MLSys is a better venue ? Again, I'm not sure.
> Although our work does not introduce a new GAN or diffusion method, we believe our paper offers a degree of novelty because the question of a fair comparison between GAN and diffusion-based super-resolution remains unexplored. Our primary finding disproves the common belief that GAN-based super-resolution is inferior to diffusion-based approaches. It is important to note that we employed the same straightforward architecture for both paradigms without implementing any GAN-specific or diffusion-specific modifications.
>
> While we agree with the reviewer and thanks for the suggestion that MLSys is suitable venue, we believe that the problem discussed may be of interest for wide audience.
>
> > Even though the experiment setting is quite fair, GAN-based models actually have one additional pretraining stage whereas diffusion model has to be trained from scratch, which could be a reason why diffusion-based model lacks behind GAN-based ones.
>
> We want to highlight that in GAN-based super-resolution (SR), an initial L1 pretraining phase is necessary. This stage is quite similar to diffusion model training, with the difference being that the target is the ground truth high-resolution image instead of applied noise. Despite involving two stages, GAN-based training demands significantly fewer resources than diffusion models. Both L1 pretraining and adversarial training typically require around 100,000 iterations, whereas diffusion models reach their best results only after 1 million iterations. Our findings suggest that the GAN-based approach is more efficient in terms of training.
>
> > The authors do not report how the GAN-based model perform without L1 pretraining stage, both qualitatively and quantitatively. Also, claiming that training GAN-based models for ISR does not face instability issues is quite a bold claim because without pretraining on L1 loss, I can imagine that it could be really unstable.
>
> In our preliminary experiments we had an experiment without L1 pretraining and it has shown poor performance.
> However, we believe that additional of L1 stage doesn’t significantly complicate the setup as it is merely a sort of warmup phase adopted in many other applications (for instance, large language model pretraining).

---

> ### Author Response · Authors · 2024-11-25
>
> **Weaknesses (part2)**
>
> > Though the paper highlights that both GANs and diffusion models benefit from scaling, it does not investigate how these models scale in terms of data. Like, how much data that GAN-based model starts to outperform diffusion ones. This could help to strengthen the work.
>
> Following your suggestion, we explored how scaling data actually influences models’ performance.Specifically, we took [18k, 180k, 1.8M] subsets from the original large training dataset and trained both GAN-based and Diffusion-based models for the same number of steps. Then we computed IQA-metrics and conducted Side-by-Side comparison between the models trained on whole dataset and smaller subsets, and, finally, between corresponding GANs and diffusion models.
>
> Table 1. Side-by-side comparison between diffusion models trained on different amount of data.
> |     	| Diff-0.018M vs Diff-18M  | Diff-0.18M vs Diff-18M |  Diff-1.8M vs Diff-18M |
> |---------|-------------------------------|-----|-----|
> | Win 	|0.167 | 0.125 | 0.163 |
> | Lose	| 0.583 | 0.242 | 0.283 |
> | Tie| 0.250 | 0.633| 0.554  |
> | p-value | 0.000 | 0.08 | 0.07|
>
> Table 2. Side-by-side comparison between GAN models trained on different amount of data.
> |     	| GAN-0.018M vs GAN-18M  | GAN-0.18M vs GAN-18M | GAN-1.8M vs GAN-18M |
> |---------|-------------------------------|-----|-----|
> | Win 	|  0.108| 0.250 | 0.142 |
> | Lose	| 0.100| 0.188 |0.183 |
> | Tie|  0.792| 0.563|0.675 |
> | p-value | 0.95 | 0.37 |0.56 |
>
>
> |     	| Diff-0.018M vs GAN-0.018M | Diff-0.18M vs GAN-0.18M |Diff-1.8M vs  GAN-1.8M |
> |---------|-------------------------------|-----|-----|
> | Win 	|  0.029| 0.017 | 0.008 |
> | Lose	| 0.838| 0.750 |0.775 |
> | Tie|  0.133| 0.233|0.217 |
> | p-value | 0.00 | 0.00 |0.00 |
>
> Table 3. IQA metrics for GAN models trained on different amount of data.
> |      	| Diff-0.018M  | Diff-0.18M| Diff-1.8M| Diff-18M |
> |----------|---------|---------|---|---|
> | PSNR 	| 24.83	|   25.33	|25.02   | 26.67 |
> | SSIM 	| 0.665 |  0.697	| 0.689| 0.748|
> | LPIPS	| 0.351	| 0.328  	|0.330  | 0.253 |
> | CLIP-IQA | 0.620 	|  0.602 | 0.647 | 0.719 |
>
> Table 4. IQA metrics for diffusion models trained on different amount of data.
> |      	| GAN-0.018M  | GAN-0.18M| GAN-1.8M| GAN-18M |
> |----------|---------|---------|---|---|
> | PSNR 	| 25.538	|   26.366 |25.811   | 26.01 |
> | SSIM 	| 0.747 |  0.770	|0.763 | 0.770|
> | LPIPS	| 0.241	| 0.233  	|0.237  | 0.208 |
> | CLIP-IQA | 0.749	|  0.758 | 0.769 | 0.826 |
>
> Remarkably, the performance of GAN-based SR saturates quickly with respect to the amount of training data.
> We observed no significant difference in terms of standard SR metrics (PSNR/SSIM/LPIPS) between the model trained on the smallest amount of data (i.e 18k samples) vs the one trained on 18M images . Only CLIP-IQA shows small improvement. However, Side-by-Side comparison between different GAN-base SR models shows equality with respect to human assessment for models trained on various amounts of data.
>
> At the same time, the Diffusion model appears to be more sensitive to data. Whereas, models trained on 180k, 1.8M match the model trained on large dataset in terms of quality, the one trained on the smallest number of samples appears to be statistically worse. This finding suggests that Diffusion-based SR is likely to be more “data-hungry” compared to GAN-based SR.
>
> Therefore, we conclude that GAN performs better than diffusion regardless of the amount of training data.

---

> ### Author Response · Authors · 2024-11-25
>
> **Questions**
>
> > Is it possible to do some "pretraining" for diffusion-based models similar to the pretraining on L1 for Gan-based method ? So that it can be more fair in evaluation.
>
> We are unaware of any sort of diffusion “pretraining” as the model is directly trained to produce images from the target distribution. At the same time, Diffusion model can be used as an initialization for GAN-based approach [1]. As discussed in the text, in our study we compare between the most standard GAN-based and Diffusion-based SR training protocol.
>
> > As I mentioned above in Weaknesses section (bullet point 4), I would like to see how data scaling affect the performance of both GAN-based and diffusion-based ISR models.
>
> We provide the results of the data-scaling experiments in the response above (see `Weaknesses (part2)`).
>
> > I know it's a bit infeasible but showing comparison on other restoration tasks like image dehazing, deblurring, .. to see if the same phenonmena happens can be really valuable and strengthen the work.
>
> We agree with the reviewer, that such comparison would be valuable and interesting for the research community. However, we leave this question for future work.
>
> References
> ---
>
> [1] Xie, Rui, et al. "Addsr: Accelerating diffusion-based blind super-resolution with adversarial diffusion distillation." arXiv preprint arXiv:2404.01717 (2024).

---

> ### Comment · Reviewer_cAL4 · 2024-11-26
> **Regarding to the Rebuttal**
>
> Thank the author for the rebuttal and here is my comment for the rebuttal:
>
> 1. I thanks that the authors point out that GAN-based model is actually more efficient than diffusion-based even with an additional pretraining stage. The author should rewrite to highlight and emphasize this, this is a strong point but I missed it and I'm sorry for this.
>
> 2. It is interesting and surprising that diffusion-based approach is more sensitive to data as it is common believe that diffusion models are more scalable. This results should be added in the main paper to strengthen the paper more.
>
> 3. Again this is a minor point as it is really infeasible to run the experiments but if possible, the authors should add comparison for other tasks  image dehazing, deblurring, .. in future version/work. This can really boost the value of the work.
>
> However, the rebuttal is still quite not enough for me, so I only increase **my score to 5**.

---

### Official Review · Reviewer_sLAS · 2024-10-27

**Soundness:** 2
**Presentation:** 2
**Contribution:** 2
**Rating:** 6
**Confidence:** 4

**Summary:**

This study challenges the assumption that diffusion-based models inherently outperform GANs in Image Super Resolution (ISR). Noting that diffusion-based ISR models often use larger networks and longer training than GANs, the authors investigate whether these performance gains come from model design or simply increased resources. By controlling for architecture, model size, dataset, and computational budget, they find that GAN-based models can achieve comparable or even superior results. They also examine the influence of factors like text conditioning and augmentation, analyzing their effects on ISR performance and related tasks.

**Strengths:**

+ Conducting a comparison of GAN and Diffusion-based approaches for Super-Resolution with the same computational resource can provide good insight for the community.
+ The finding that given the same model size, GAN matches the performance of quality with the diffusion-based method is interesting.

**Weaknesses:**

**Concerns**
I believe that the paper needs to have a thorough clarification. Specifically:

+ Claiming that GAN-based and Diffusion-based approaches give the comparison if using the same number of parameters might be relatively strong. It needs very careful investigations and evaluation. Because, if they give comparable performance, the community has no reason to use diffusion with much more cost for both training (much longer) and inference (much more sampling steps). A comparison with the same setup on some widely-used common dataset benchmarks at first might provide some insights and support rather than just collecting some custom massive datasets.

+ Conducting experiments on extremely huge data, i.e. 17 million images is a very high cost. The author could provide a comparison from a small to a larger number of data in their collected data to see the differences between GAN-based and Diffusion-based methods. For example, 100k, 1M, 2M, 5M, 17M, etc pickup of some of these settings might be reasonable to see if the results/findings are consistent. In practice and research, the number of images for the study is often not too huge up to 17M.

+ Were the methods in Table 1, and Table 4 "SUPIR, RealESRGAN, DiffBIR, ResShift" trained on the same dataset as the Diff (ours) and GAN (ours)? Also in these tables, it should be better to clearly state which one is GAN-based and diffusion-based would greatly improve readability.

+ Table 1 and Table 8 show that ResShift with about just 1/4 parameters (174M) already outperformed the GAN (ours) and Diffusion (ours) 614 and 630M on PSNR and SSIM. This may raise a question of whether scaling more can bring up the performance or not, which is contradicting as concerned in the paper doubts the performance gain that comes from scaling up model size.

+ Figure 1 and Figure 4 are almost the same and seem to be redundant with no more information added.

**Other suggestions**
+ For the whole paper, the current form seems to use all \cite{} making it very messy for all references. I think the use of \citep{} in latex would produce a more correct presentation of citations for many parts of the paper. Using \cite{} for cases where the citing author is subject (S), but \citep{} for other cases when referring to the paper.
+ Text in figures presented in the paper is too small, e.g. Figure 2, figure 3, figure 5.

**Questions:**

Q1. The detail of "we did not encounter any difficulties with optimization" --> What can be the reasons for that nice success? Since many works and practices confirm that training GAN is very unstable and mode collapse is a well-known problem of GAN.

Q2. An experiment for the diffusion-based method in this study took 1 month to get the checkpoint, didn't it? Here many experiments were conducted for diffusion, how much time (months) it is estimated to take to complete all of the reported training? This is to provide some information for reproducibility for the community.

---

> ### Author Response · Authors · 2024-11-25
>
> We appreciate the reviewer’s insightful comments and suggestions. Questions and concerns are addressed below:
>
> **Weaknesses (part1)**
>
> > Claiming that GAN-based and Diffusion-based approaches give the comparison if using the same number of parameters might be relatively strong. It needs very careful investigations and evaluation. Because, if they give comparable performance, the community has no reason to use diffusion with much more cost for both training (much longer) and inference (much more sampling steps). A comparison with the same setup on some widely-used common dataset benchmarks at first might provide some insights and support rather than just collecting some custom massive datasets.
>
> Our study emphasizes the superior performance of GANs over Diffusion models in the traditional Image Super Resolution task, challenging common assumptions and demonstrating this under controlled conditions. To provide clearer insights, as per your suggestion, we trained both our diffusion model and GAN using a random sample of 18,000 images from the open-source LSDIR dataset. We then calculated relevant metrics and conducted a side-by-side comparison of the final models.
>
> Table 1. Diffusion vs GAN on LSDIR datasets.
>  |     	| Diff-LSDIR vs GAN-LSDIR  |
>  |---------|-------------------------------|
>  | Win 	|  0.046|
>   | Lose	| 0.571|
>   | Tie|  0.383|
>   | p-value | 0.000|
>
> Table 2. IQA-metrics for SR models trained on LSDIR.
>
> | | Diff-LSDIR  | GAN-LSDIR |
> |----------|---------|---------|
> | PSNR 	| 24.29 	| 26.43  	|
> | SSIM 	| 0.646 | 0.772 	|
> | LPIPS	| 0.362 	| 0.232  	|
> | CLIP-IQA |   0.651	|   0.757 |
>
>  It is clear from the results above that the main claim is still valid.

---

> ### Author Response · Authors · 2024-11-25
>
> **Weaknesses (part 2)**
>
> >  Conducting experiments on extremely huge data, i.e. 17 million images is a very high cost. The author could provide a comparison from a small to a larger number of data in their collected data to see the differences between GAN-based and Diffusion-based methods. For example, 100k, 1M, 2M, 5M, 17M, etc pickup of some of these settings might be reasonable to see if the results/findings are consistent. In practice and research, the number of images for the study is often not too huge up to 17M.
>
> According to your suggestion we performed experiments with smaller amount of samples both for text-unconditional GAN-based and Diffusion-based SR. Specifically, we took [18k, 180k, 1.8M] subsets from the original large training dataset and trained both GAN-based and Diffusion-based models for the same number of steps. Then we computed IQA-metrics and conducted Side-by-Side comparison between the models trained on whole dataset and smaller subsets.
>
> Table 3. Side-by-side comparison between diffusion models trained on different amount of data.
> |     	| Diff-0.018M vs Diff-18M  | Diff-0.18M vs Diff-18M |  Diff-1.8M vs Diff-18M |
> |---------|-------------------------------|-----|-----|
> | Win 	|0.167 | 0.125 | 0.163 |
> | Lose	| 0.583 | 0.242 | 0.283 |
> | Tie| 0.250 | 0.633| 0.554  |
> | p-value | 0.000 | 0.08 | 0.07|
>
> Table 4. Side-by-side comparison between GAN models trained on different amount of data.
> |     	| GAN-0.018M vs GAN-18M  | GAN-0.18M vs GAN-18M | GAN-1.8M vs GAN-18M |
> |---------|-------------------------------|-----|-----|
> | Win 	|  0.108| 0.250 | 0.142 |
> | Lose	| 0.100| 0.188 |0.183 |
> | Tie|  0.792| 0.563|0.675 |
> | p-value | 0.95 | 0.37 |0.56 |
>
> Table 5. IQA metrics for GAN models trained on different amount of data.
> |      	| Diff-0.018M  | Diff-0.18M| Diff-1.8M| Diff-18M |
> |----------|---------|---------|---|---|
> | PSNR 	| 24.83	|   25.33	|25.02   | 26.67 |
> | SSIM 	| 0.665 |  0.697	| 0.689| 0.748|
> | LPIPS	| 0.351	| 0.328  	|0.330  | 0.253 |
> | CLIP-IQA | 0.620 	|  0.602 | 0.647 | 0.719 |
>
> Table 6. IQA metrics for diffusion models trained on different amount of data.
> |      	| GAN-0.018M  | GAN-0.18M| GAN-1.8M| GAN-18M |
> |----------|---------|---------|---|---|
> | PSNR 	| 25.538	|   26.366 |25.811   | 26.01 |
> | SSIM 	| 0.747 |  0.770	|0.763 | 0.770|
> | LPIPS	| 0.241	| 0.233  	|0.237  | 0.208 |
> | CLIP-IQA | 0.749	|  0.758 | 0.769 | 0.826 |
>
> Remarkably, the performance of GAN-based SR saturates quickly with respect to the amount of training data.
> We observed no significant difference in terms of standard SR metrics (PSNR/SSIM/LPIPS) between the model trained on the smallest amount of data (i.e 18k samples) vs the one trained on 17M images rendering. Only CLIP-IQA shows small improvement. At the same time, Side-by-Side comparison between different GAN-base SR models shows equality with respect to human assessment for models trained on various amounts of data.
>
> At the same time, the Diffusion model appears to be more sensitive to data. Whereas, models trained on 180k, 1.8M match the model trained on large dataset in terms of quality, the one trained on the smallest number of samples appears to be statistically worse. This finding suggests that Diffusion-based SR is likely to be more “data-hungry” compared to GAN-based SR.

---

> ### Author Response · Authors · 2024-11-25
>
> **Weaknesses (part3)**
>
> > Were the methods in Table 1, and Table 4 "SUPIR, RealESRGAN, DiffBIR, ResShift" trained on the same dataset as the Diff (ours) and GAN (ours)? Also in these tables, it should be better to clearly state which one is GAN-based and diffusion-based would greatly improve readability.
>
> All open-source models used in this work as baselines were taken as is from their original repositories.
> Each model was trained on a different dataset with specific training protocol. For instance,  SUPIR  was trained on an exceptionally large proprietary dataset, whereas RealESRGAN used relatively small mix of DIV2k/OST data. We note that we neither aim to provide comparison in a controlled setting with all related work in the field, nor to establish new SOTA. The main purpose of providing comparison with these models is to show that protocol used in our work is performant enough to be practically interesting.
>
> Following the reviewer’s suggestion we added the markers indicating the type of Super Resolution model to improve readability.
>
> > Table 1 and Table 8 show that ResShift with about just 1/4 parameters (174M) already outperformed the GAN (ours) and Diffusion (ours) 614 and 630M on PSNR and SSIM. This may raise a question of whether scaling more can bring up the performance or not, which is contradicting as concerned in the paper doubts the performance gain that comes from scaling up model size.
>
> While classic full-reference metrics such as PSNR or SSIM may show advantage of ResShift model, it is known from numerous research papers on IQA [5, 6] that these metrics might not be representative for evaluation of super-resolution models’ performance. Keeping that in mind, we provide both human evaluation and non-reference CLIP-IQA metric for a better comparison. According to them (see Tables 1, 2, 4, 5) bigger models perform better on average than smaller models (particularly, better than ResShift).
>
> >  Figure 1 and Figure 4 are almost the same and seem to be redundant with no more information added.
>
> We believe that Table 1 and Table 4 are both essential as they show models’ performance in different settings: Table 1 reflects performance on classical ISR task where the objective is inversion of downscaling operation without any additional degradations, while Table 4 shows performance on the blind ISR task (which implies that models were trained with complex degradation pipeline).

---

> ### Author Response · Authors · 2024-11-25
>
> **Questions**
>
>  > The detail of "we did not encounter any difficulties with optimization" --> What can be the reasons for that nice success? Since many works and practices confirm that training GAN is very unstable and mode collapse is a well-known problem of GAN.
>
> We note that works acknowledging the difficulties of GAN optimization come from the problem of unconditional and conditional image generation, where there is a large freedom in the number of possible outputs. Most of the literature in the Image Super-Resolution [2,3,4] adopts a pretty simple GAN training protocol and doesn’t mention the need for extensive tuning. We would like to emphasize that the Image Super Resolution task is very different from image generation setup as the low-resolution image serves already as a strong prior and the aim of adversarial loss is to facilitate production of sharp details and textures.
>
>  > An experiment for the diffusion-based method in this study took 1 month to get the checkpoint, didn't it? Here many experiments were conducted for diffusion, how much time (months) it is estimated to take to complete all of the reported training? This is to provide some information for reproducibility for the community.
>
> We were training each of our diffusion models on 16 NVIDIA Tesla A100 GPU with 80GB of VRAM, physical batch size 32, PyTorch 2.4.0, and CUDA 12.4. It took 1.125 training iterations per second in average, so, excluding any server crashes or other technical problems, training of diffusion models takes approximately 11 days until convergence.
>
> References
> ---
> [1] Li, Yawei, et al. "Lsdir: A large scale dataset for image restoration." Proceedings of the IEEE/CVF Conference on Computer Vision and Pattern Recognition. 2023.
>
> [2] Ledig, Christian, et al. "Photo-realistic single image super-resolution using a generative adversarial network." Proceedings of the IEEE conference on computer vision and pattern recognition. 2017.
>
> [3] Wang, Xintao, et al. "Esrgan: Enhanced super-resolution generative adversarial networks." Proceedings of the European conference on computer vision (ECCV) workshops. 2018.
>
> [4] Wang, Xintao, et al. "Real-esrgan: Training real-world blind super-resolution with pure synthetic data." Proceedings of the IEEE/CVF international conference on computer vision. 2021
>
> [5] Wang, Zhihao, Jian Chen, and Steven CH Hoi. "Deep learning for image super-resolution: A survey." IEEE transactions on pattern analysis and machine intelligence 43.10 (2020): 3365-3387.
>
> [6] Yu, Fanghua, et al. "Scaling up to excellence: Practicing model scaling for photo-realistic image restoration in the wild." Proceedings of the IEEE/CVF Conference on Computer Vision and Pattern Recognition. 2024.

---

> > ### Comment · Reviewer_sLAS · 2024-12-01
> >
> > The rebuttal has addressed my concerns to some extent. While the contribution leans more toward empirical engineering, the paper may offer valuable insights that could benefit the deep-learning community. Therefore, I am raising my score.

---

### Official Review · Reviewer_euHF · 2024-11-04

**Soundness:** 3
**Presentation:** 2
**Contribution:** 2
**Rating:** 6
**Confidence:** 4

**Summary:**

This paper systematically compares GANs and diffusion models for image super-resolution (ISR) under controlled, comparable conditions. The findings reveal that GANs, when trained with similar protocols, can match the quality of diffusion models while offering practical advantages, such as faster training and single-step inference. This study highlights key trade-offs between GAN and diffusion approaches, contributing valuable insights into their performance and efficiency in ISR tasks.

**Strengths:**

Overall, this type of work should be appreciated as it probes deeper into what the differences in paradigms are when it comes to the SR task. The authors ensure that the setup for both paradigms is as comparable as possible through the architecture, datasets, etc. In general, the writing is good but some parts were confusing.

**Weaknesses:**

Because their experimental results were sometimes in favor of diffusion, sometimes in favor of GAN, and sometimes no difference was found, I have two suggestions that would significantly improve this paper.  First, the authors should taxonomize and explain to the reader when one paradigm should be preferred and in which scenarios. Otherwise, this type of work does not help us improve actual application performance. Second, for the authors to actually be able to make such claims, they need to use more fine-grained evaluation, e.g. the model outputs (SR) images should be used for a specific task like digit recognition, segmentation, etc. However, just comparing single valued PSNR, SSIM, LPIPS etc does not really tell us where these models are outperforming each other. That is also very evident from the qualitative results where within the same figure, both paradigms *visually outperform* each other.

----not necessarily weaknesses----
Terminology: In Line 49, the term “fairness” might be misleading in this context. Instead, a term like “controlled conditions” or “standardized experimental setup” could better communicate the need for consistent variables, such as dataset size and model complexity, in comparing results.
-  In the related works section, the authors mention conflicting findings about whether text conditioning improves diffusion-based ISR. However, it’s unclear why these differences exist or what insights the current paper offers on this topic. A more thorough discussion or stance on this issue could add depth and relevance.
- Moving the “variations of SR” subsection earlier in the paper would help readers understand the exact ISR task being investigated, providing important context before diving into the model comparisons.
- In Line 130, “given a reference on training” is unclear.

- A significant limitation is the use of proprietary models and datasets, making it difficult for others to replicate the experiments. For instance, the use of “internal foundation model” and a “proprietary dataset of 17 million…” lacks important detail.  Will this dataset be released?

Figure Captions and Clarity:
Figures 1 and 4: These figures would benefit from more descriptive captions, highlighting key differences and the main takeaways.
Figure 4: The meaning of the green and grey indicators should be further clarified, as well as the criteria used to define convergence in the caption. I’m aware it is in the text.
Figures 2 and 5: Why do these not include the original HR?
Figure 3: This figure is hard to interpret because it’s unclear what exact quantity or metric is being reported. Should be added to the caption. The corresponding section is also difficult to understand.

**Questions:**

“We note that we use semantic image-level captions. Global-level information appears to be not very useful for the SR task” What is the difference between image-level and global-level. Can you give an example? This is unclear.

I have never seen “p-value on SbS comparison” as a way to evaluate using human judgment. Why not just do this quantitatively and threshold the difference between checkpoints? The current way of stopping and evaluating seems incredibly arbitrary and subjective. Can the authors share a few works that use this paradigm?  as I have never encountered it.
E.g. also in “ We conduct an SbS comparison between text-conditioned and unconditional models for both paradigms at all stages of training”
How exactly are the GAN models conditioned on the text?

One major issue is that because of the nature of the SR problem (ill-posed, many to many mapping etc) the types of errors SR models make can inherently invalidate the correctness of the image, e.g. numbers get blurred. It would make the paper significantly stronger if the authors can dig deeper into the entire performance of the SR models for specific tasks as they pertain to downstream tasks and make the evaluation more human interpretable. Right now it is being collapsed into a single number and some qualitative examples. It’s really challenging to make use of the findings in the paper for downstream research/applications. This could significantly enhance the contributions of this work (e.g. by answering *when should one use one paradigm over the other*)
But really what are the types of error differences between the two?
E.g. figure 9 in appendix - the digits or letters.
E.g. figure 12 in appendix - sometimes diffBIR and real esrgan , the results are switched, one does better than the other
We are not really getting conclusive results. Instead mixed findings. It would be great if the authors can taxonimize and dig deeper into when we should use GAN over Diffusion based SR models.
Line 1505 - figure caption “high frequency”
Not obvious to me why in section G, “G SUPER RESOLUTION OF SYNTHETIC IMAGES” the authors used those datasets to test for OOD? Why not use real data not synthetic data? Second, what if the synthetic data generated by diffusion models (e.g. SDXL as mentioned in the paper) may actually produce a distribution that is closer to that of the diffusion based SR model, thereby giving the diffusion based model a sort of advantage?
Table 1 What is the dataset being used? Authors should mention this in the caption of the table.

---

> ### Author Response · Authors · 2024-11-25
>
> We thank the reviewer for a thoughtful and meticulous review. We address the concerns below.
>
> **Weaknesses**
>
> > Because their experimental results were sometimes in favor of diffusion, sometimes in favor of GAN, and sometimes no difference was found, I have two suggestions that would significantly improve this paper. First, the authors should taxonomize and explain to the reader when one paradigm should be preferred and in which scenarios. Otherwise, this type of work does not help us improve actual application performance. Second, for the authors to actually be able to make such claims, they need to use more fine-grained evaluation, e.g. the model outputs (SR) images should be used for a specific task like digit recognition, segmentation, etc. However, just comparing single valued PSNR, SSIM, LPIPS etc does not really tell us where these models are outperforming each other. That is also very evident from the qualitative results where within the same figure, both paradigms visually outperform each other.
>
> We present both classical full-reference and no-reference metrics in line with the standard evaluation protocol commonly used in Super Resolution research. Although evaluating performance on downstream tasks is an interesting approach, it is not widely adopted and may not accurately represent performance in the Image Super Resolution (ISR) task, which is the primary focus of our study. We primarily rely on Human Preference Studies, as elaborated in `Section 4.3 `of the main text, with results detailed in `Tables 2` and `5`. Given that we gather data from numerous assessors for each comparison between two models, we believe this methodology provides a reliable measurement of each model's performance. Moreover, focus on specialized downstream tasks may narrow down the potential applicability of generated images outside of the particular application of interest while the main goal of ISR is to produce general purpose image enhancement.
>
> > ---not necessarily weaknesses---- Terminology: In Line 49, the term “fairness” might be misleading in this context. Instead, a term like “controlled conditions” or “standardized experimental setup” could better communicate the need for consistent variables, such as dataset size and model complexity, in comparing results.
>
> Thank you for the suggestion, we edited this place in the text..
>
> > In the related works section, the authors mention conflicting findings about whether text conditioning improves diffusion-based ISR. However, it’s unclear why these differences exist or what insights the current paper offers on this topic. A more thorough discussion or stance on this issue could add depth and relevance.
>
> In this paragraph, we highlight several recent notable studies that explore the use of text conditioning, with some studies reporting an improvement in quality while others do not. However, the referenced works lack detailed ablation analyses and comparative studies on this subject. In our paper, we conduct a thorough ablation study of this design choice by examining two different text encoder options. Through a user preference study, we demonstrate that text conditioning, under protocols similar to those used in the referenced works, does not impact Super Resolution (SR) performance.
>
> > in Line 130, “given a reference on training” is unclear.
>
> This implies that during the training process, the GAN receives a low-resolution image along with its **high-resolution reference** to compute the loss.
>
> > A significant limitation is the use of proprietary models and datasets, making it difficult for others to replicate the experiments. For instance, the use of “internal foundation model” and a “proprietary dataset of 17 million…” lacks important detail. Will this dataset be released?
>
> The Image Super Resolution models and the inference code will be made available following the acceptance of our work. A detailed description of the data collection process can be found in `Appendix F`. Regrettably, we are unable to release the dataset due to our organization's internal policy.

---

> ### Author Response · Authors · 2024-11-25
>
> **Questions** (part1)
>
> > “We note that we use semantic image-level captions. Global-level information appears to be not very useful for the SR task” What is the difference between image-level and global-level. Can you give an example? This is unclear.
>
> We apologize for any confusion caused by our terminology. The terms "image-level" and "global-level" are used interchangeably and refer to the same concept. An image-level caption typically provides a brief description of the scene, such as "A photo of the Eiffel Tower at sunset." Our objective was to demonstrate that using captions with general descriptive information about the image (i.e., "image-level" or "global-level" information) does not enhance the model's performance. We use these terms to underline the conceptual difference between general purpose image cations that could be used for text-to-image models training and ISR-specific image captions e.g. the ones proposed in SUPIR, which are not easily accessible in general.
>
> >  have never seen “p-value on SbS comparison” as a way to evaluate using human judgment. Why not just do this quantitatively and threshold the difference between checkpoints? The current way of stopping and evaluating seems incredibly arbitrary and subjective. Can the authors share a few works that use this paradigm? as I have never encountered it. E.g. also in “
>
> The ultimate objective of the Image Super Resolution task is to produce visually appealing images with sharp edges and fine details. We propose that early stopping based on human preferences—specifically when assessors cannot distinguish between two consecutive training snapshots—is a reasonable strategy for monitoring convergence. To the best of our knowledge, this approach is novel and has not been utilized in prior work. However, the proposed protocol is based on a preference study that can be treated as an experimental validation of a statistical hypothesis.  Statistical tests are standard tools for assessing the significance of user preference studies. According to conventional protocols [4], if the p-value of a test is less than 0.05, we consider the difference between the two models to be statistically significant, meaning it is very unlikely to have occurred by chance.
>
> > We conduct an SbS comparison between text-conditioned and unconditional models for both paradigms at all stages of training. How exactly are the GAN models conditioned on the text?
>
> In both Diffusion and GAN models, text conditioning is implemented identically through Cross-Attention between image and text tokens together with scale-shift modulation in Convolutional Residual Blocks. This occurs at the lowest resolution within the UNet encoder, decoder, and middle block, as detailed in the "Text Conditioning" section of Section 3. This design choice is consistent with the approaches used in the SR3 [5]  and Imagen [3].
>
> >  E.g. figure 9 in appendix - the digits or letters. E.g. figure 12 in appendix - sometimes diffBIR and real esrgan , the results are switched, one does better than the other We are not really getting conclusive results. Instead mixed findings. It would be great if the authors can taxonimize and dig deeper into when we should use GAN over Diffusion based SR models
>
> To ensure a comprehensive evaluation of the model's performance, we conducted assessments on a dataset of 244 images, with multiple votes collected for each image. While individual examples may not fully illustrate the differences between methods, our findings indicate that GAN-based super resolution (SR) outperforms diffusion-based SR under the controlled training protocol. Our GAN-based model consistently surpasses literature baselines in the classical SR task, though it may underperform in super resolution with degraded images, such as when compared to SUPIR [6], DiffBIR [7].
>
> Generally, our results suggest that GAN-based SR is preferable in most situations, as it is faster to train, delivers superior super resolution performance, and offers significantly quicker inference (using a single-step super resolution instead of iterative denoising). However, both paradigms exhibit specific artifacts, as discussed in Section 4.6.

---

> ### Author Response · Authors · 2024-11-25
>
> **Questions** (part2)
>
> > Not obvious to me why in section G, “G SUPER RESOLUTION OF SYNTHETIC IMAGES” the authors used those datasets to test for OOD? Why not use real data not synthetic data?
>
> We used these datasets as an interesting case study. Cascaded diffusion models [1, 2, 3], which were popular some time ago [1] in the text-2-generation comprise base generator and single or multiple Super Resolution stages to generate high-resolution images. Moreover, modern Latent Diffusion models also produce images that could be further enhanced with a stand alone SR model. We show that SR models from our papers work reasonably well on top of the generations of cascaded (YaART)[1] and latent (SDXL) diffusion models and may serve for generation of 4k images. As there is no ground truth reference available, we have not reported any image quality assessment (IQA) metrics.
>
> > Second, what if the synthetic data generated by diffusion models (e.g. SDXL as mentioned in the paper) may actually produce a distribution that is closer to that of the diffusion based SR model, thereby giving the diffusion based model a sort of advantage?
>
> Both diffusion-based and GAN-based super resolution (SR) models are trained on real data, making the outputs generated by diffusion models out-of-distribution (OOD) for both approaches. Although one might assume that diffusion models would be more robust to distribution shifts, both types of models actually perform well on these OOD inputs. Typically, GAN-based models produce images with sharper edges and finer details, similar to their results on real data.
>
> > Table 1 What is the dataset being used? Authors should mention this in the caption of the table.
>
> We used the same dataset for all evaluations (both with respect to IQA metrics and user preference study). Details about the dataset used are provided in paragraph “Evaluation datasets”. The dataset involves samples from RealSR, DRealSR, which are standard in SR literature, as well as additional samples from the web to provide more detailed and extensive comparison between models.
>
> References
> ---
> [1] Kastryulin, Sergey, et al. "YaART: Yet Another ART Rendering Technology." arXiv preprint arXiv:2404.05666 (2024).
>
> [2] Ho, Jonathan, et al. "Cascaded diffusion models for high fidelity image generation." Journal of Machine Learning Research 23.47 (2022): 1-33.
>
> [3] Saharia, Chitwan, et al. "Photorealistic text-to-image diffusion models with deep language understanding." Advances in neural information processing systems 35 (2022): 36479-36494.
>
> [4] Wasserstein, Ronald L., and Nicole A. Lazar. "The ASA statement on p-values: context, process, and purpose." The American Statistician 70.2 (2016): 129-133.
>
> [5] Saharia, Chitwan, et al. "Image super-resolution via iterative refinement." IEEE transactions on pattern analysis and machine intelligence 45.4 (2022): 4713-4726.
>
> [6] Yu, Fanghua, et al. "Scaling up to excellence: Practicing model scaling for photo-realistic image restoration in the wild." Proceedings of the IEEE/CVF Conference on Computer Vision and Pattern Recognition. 2024.
>
> [7] Lin, Xinqi, et al. "Diffbir: Towards blind image restoration with generative diffusion prior." arXiv preprint arXiv:2308.15070 (2023).

---

> > ### Comment · Reviewer_euHF · 2024-11-26
> >
> > Thank you for addressing all of my concerns individually. I also took the time to read other reviews, and along with your thorough responses, they have prompted me to increase my rating from 5 to 6.

---

### Meta-Review · Area_Chair_kB6X · 2024-12-17

**Metareview:**

This work focuses on the comparison of GAN and diffusion model in the task of super-resolution. Specifically, GANs and diffusion models with similar parameters are trained, and their performance is observed. The findings of the paper are that GANs achieve comparable results to diffusion models, but with the merits of faster convergence and single-step inference. Reviewers generally agree that the findings in this work are interesting, and the AC shares the same view.

This paper receives mixed ratings of (5, 6, 6, 6). After reading the paper, review, and rebuttal, the AC has concerns about the comprehensiveness of the analysis. For example:

1. This paper mainly focuses on one design choice for each model, and a rather limited training paradigms. For example, the conclusion could be different for different architecture (e.g., UNet vs DiT) and pre-training scheme (e.g., text-to-image pre-training). While the current findings is somewhat surprising and interesting, more thorough studies should be conducted.

2. The studies of parameter count can be more comprehensive. For example, while it is shown that GANs achieve comparable performance to diffusion models with around 600M - 700M parameters, it is unclear the same conclusion can be reached for say fewer or more parameters. This study is important to know the limits and potential of both models.

As an analytical paper, it is recommended to include more studies and comparison between the two paradigms in the future version. Based on the concerns, the AC would recommend a rejection.

**Additional Comments On Reviewer Discussion:**

The reviewers generally raise technical details about the studies (e.g., different dataset sizes), and the authors are able to resolve a large portion of them. While the AC appreciates the efforts of the authors and agrees that the problem is worth discussing, the AC thinks that the analysis in its current form cannot lead to a convincing conclusion, and hence recommends a rejection.

---

### Decision · Program_Chairs · 2025-01-22

Reject